# Improving LoRA in Privacy-preserving Federated Learning

**Youbang Sun**[*]
Dept. of Mechanical & Industrial Engineering
Northeastern University
{sun.youb}@northeastern.edu

**Zitao Li, Yaliang Li & Bolin Ding**
Alibaba Group
{zitao.l, yaliang.li,
 bolin.ding}@alibaba-inc.com

## Abstract

Low-rank adaptation (LoRA) is one of the most popular task-specific parameter-efficient fine-tuning (PEFT) methods on pre-trained language models for its good performance and computational efficiency. LoRA injects a product of two trainable rank decomposition matrices over the top of each frozen pre-trained model module. However, when applied in the setting of privacy-preserving federated learning (FL), LoRA may become unstable due to the following facts: 1) the effects of data heterogeneity and multi-step local updates are non-negligible, 2) additive noise enforced on updating gradients to guarantee differential privacy (DP) can be amplified and 3) the final performance is susceptible to hyper-parameters. A key factor leading to these phenomena is the discordance between jointly optimizing the two low-rank matrices by local clients and separately aggregating them by the central server. Thus, this paper proposes an efficient and effective version of LoRA, **F**ederated **F**reeze **A LoRA** (FFA-LoRA), to alleviate these challenges and further halve the communication cost of federated fine-tuning LLMs. The core idea of FFA-LoRA is to fix the randomly initialized non-zero matrices and only fine-tune the zero-initialized matrices. Compared to LoRA, FFA-LoRA is motivated by practical and theoretical benefits in privacy-preserved FL. Our experiments demonstrate that FFA-LoRA provides more consistent performance with better computational efficiency over vanilla LoRA in various FL tasks.

## 1 Introduction

Recent years have witnessed tremendous success in the development of large language models (LLMs) (Touvron et al., 2023; OpenAI, 2023; Zhang et al., 2022a; Zeng et al., 2022). The applications of LLMs range from a versatile chatbot for different writing tasks (OpenAI) to multi-modal systems (Driess et al., 2023; Wu et al., 2023; Bommasani et al., 2021). Besides the commercialized products based on general-purpose LLMs, people can also build their customized LLMs by utilizing their task-specific data to fine-tune pre-trained LLMs (Howard & Ruder, 2018). Since modern LLMs usually contain billions of parameters, fine-tuning on all parameters has prohibitively high computational costs. As a remedy, parameter efficient fine-tuning (PEFT) approaches (Ding et al., 2023), such as Low-Rank Adaptation (LoRA) (Hu et al., 2021) have been developed and commonly adapted in many downstream tasks. PEFT methods freeze the majority of parameters in pre-trained LLMs, and perform update on a small subset of parameters. Compared to full model fine-tuning, these approaches usually offer on-par or even better performance while significantly improving computational efficiency.

In this paper, we focus on LoRA for its good performance and versatility for a wide spectrum of tasks with many variations. However, LoRA still requires sufficient training data to achieve significant improvement over the raw model. The data-limited parties can unite with others and adopt federated learning (FL) (Li et al., 2020) as the computation framework to fine-tune the model collaboratively. The parameter-efficient nature of LoRA is welcomed in FL due to its low communication costs and relatively low local computational burdens. Furthermore, if the data parties in FL (usually known as *clients* in FL) want to provably prevent local data leaking from their shared information in FL,

---

[*]Work was done while the first author Youbang Sun was an intern at Alibaba Group.

differential privacy (DP) (Dwork et al., 2006) techniques can be further employed to provide privacy guarantees.

While there are many existing research results exploring (privacy-preserved) PEFT in the central setting, the exploration on how to conduct (privacy-preserved) LoRA in the FL setting is still a premature. Directly migrating LoRA methods from the central setting and combining it with FedAvg may not achieve the best performance since other sources of interference in the (privacy-preserving) FL setting, such as noisy gradients and non-iid distribution of data in the cross-silo setting, can play important roles in the optimization process. In real-world LLM applications with privacy concerns, such as federated fine-tuning (Babakniya et al., 2023) or fine-tuning under differential privacy guarantees (Li et al., 2022), the performance of LoRA often suffers deterioration.

**Contributions.** In this paper, we identify three discordances in applying LoRA in the privacy-preserved FL setting. The first is presented as a mismatched term brought by the joint local updates and separate global aggregations on the two sets of low-rank matrices of LoRA. The second discordance is that if we employ DP-SGD as the differentially private optimizer for training, the injected noise can be amplified by the locally "semi-quadratic" nature of LoRA. Lastly, the choice of one hyper-parameter of LoRA, the scaling factor $\alpha$, can significantly affect the convergence and performance of the final model, no matter enforcing DP or not.

To resolve these discordances, we propose our solution named **F**ederated **F**reeze **A LoRA** (FFA-LoRA). FFA-LoRA freezes the non-zero initialized low-rank matrices and only perform update and aggregation on the zero-initialized matrices, only half as many parameters as LoRA. Beside FFA-LoRA's obvious effect of saving half of the communication and computational cost in FL, we also provide intuitions on why it can alleviate the three aforementioned discordances. We conduct comprehensive experiments to demonstrate the advantages of FFA-LoRA over LoRA in privacy-preserving FL, across different tasks, hyper-parameters and privacy protection levels.

We summarize our contributions as follows:

• We explore the conditions in privacy-preserved FL that are discordant with LoRA, and provide explanations on the potential reasons of this performance degradation.

• We propose a new method, FFA-LoRA, which tailors LoRA to increase its performance in these undesirable but unavoidable conditions in privacy-preserved FL.

• We conduct extensive experiments to verify that FFA-LoRA can consistently outperform LoRA.

## 2 BACKGROUND AND RELATED WORKS

**Parameter efficient fine-tuning.** The ever-increasing network size of LLMs makes them prohibitively expensive, if possible at all, to fine-tune directly. To mitigate this problem, parameter-efficient fine-tuning (PEFT) methods have been proposed. These methods introduce a small number of additional trainable parameters $\Theta$ to improve model performance and keep most of the pre-trained parameters $\Phi$ frozen. The task-specific increment $\Delta\Phi$ is then encoded into $\Delta\Theta$ with much smaller dimensions. Houlsby et al. (2019) added additional trainable neural modules named *adapters* to each layer of the network. Alternatively, prefix-tuning(Li & Liang, 2021) and prompt-tuning(Lester et al., 2021) modify the network by concatenating additional trainable dimensions to input or hidden layers of the network. Another series of works (Hu et al., 2021; Yu et al., 2021b) proposed LoRA and RGP, using low-rank matrices to approximate or re-parameterize the pre-trained weight matrices. LoRA is arguably the most popular approach among PEFT methods, it only requires tuning less than 1% of the parameters in the full fine-tune approach but achieves comparable performance in a wide range of downstream tasks. There are also works (He et al., 2021; Chavan et al., 2023) that seek to provide a generalized method that unifies these PEFT methods.

**Federated fine-tuning with LLM.** Although fine-tuned LLMs can become backbones for applications in different areas, the fine-tuning process still favors large-scale, domain-specific data. However, such domain-specific data is typically possessed by multiple parties, with each party's dataset only containing inadequate data to fine-tune models by itself. Furthermore, these parties are often prohibited from sharing such data directly with other entities. A common solution for this dilemma is federated learning (Kairouz et al., 2021), which allows a set of agents to fine-tune LLMs efficiently by sharing their local model updates without explicitly sharing their respective data. Tian et al. (2022) proposed FedBERT and performed federated pre-training on the BERT model. Different from traditional machine learning models, LLM's tremendous model size can consume significant

amount of resources for cross-party communication and require immense computation resources for local training. Many research solutions rely on the combination of PEFT with FL. There have been multiple studies of PEFT in FL in the recent years, Zhang et al. (2022b) considers PEFT in the federated setting. Recently, (Kuang et al., 2023) proposed FS-LLM, a federated framework for federated fine-tuning LLMs. It has been pointed out that data heterogeneity in FL is a challenge for PEFT algorithm (Kairouz et al., 2021; Babakniya et al., 2023).

**PEFT with differential privacy.** Although LLMs are powerful tools and offer great performance thanks to their ability to extract rich features with the transformer structure and large number of parameters, it is also well known that LLMs with large number of parameters can leak critical information contained in the training dataset (Carlini et al., 2021; Huang et al., 2022). A popular privacy notion that can provide theoretical guarantees against training data leakage from the model is differential privacy (DP) (Dwork et al., 2006).

**Definition 1** (($\epsilon, \delta$)-DP ). *A randomized algorithm $\mathcal{A}$ is ($\epsilon, \delta$)-differentially private if for any two neighboring datasets $\mathbb{D}$ and $\mathbb{D}'$, which differ in exactly a single record, and for all possible subsets $\mathbb{S} \subset \mathcal{O}$ of possible outputs of $\mathcal{A}$: $Pr[\mathcal{A}(\mathbb{D}) \in \mathbb{S}] \leq e^\epsilon Pr[\mathcal{A}(\mathbb{D}') \in \mathbb{S}] + \delta$.*

Intuitively, DP ensures that any single record cannot significantly affect the distribution of the output. With such indistinguishable output distributions, any adversary can only gain limited additional knowledge about whether a specific record is in the input data. The level of privacy is denoted by the privacy parameters ($\epsilon, \delta$), a smaller choice of ($\epsilon, \delta$) means a stronger privacy protection guarantee.

*Machine learning with DP: DP-SGD.* A classic mechanism used to ensure the published model differentially private is DP-SGD (Song et al., 2013; Abadi et al., 2016; Bassily et al., 2014). It requires a DP optimizer to privatize gradients before using them to update the model. Compared with the vanilla stochastic gradient descent (SGD) algorithm, DP-SGD has two additional operations in each iteration. It first clips per-sample gradients with a norm constraint $C$ to limit the maximum influence of any sample. Then, it adds a Gaussian noise $z \sim \mathcal{N}(0, C^2\sigma^2 I_p)$ to the sum of clipped gradients in a batch $\mathcal{B}$. Namely, $\bar{g} = \left( \sum_{i \in \mathcal{B}} \text{Clip}(\nabla f_i, C) + z \right) / |\mathcal{B}|$. Finally, this noisy sum of clipped gradients $\bar{g}$ is used to update the model. The scalar $\sigma$ is decided by privacy composition rules (Abadi et al., 2016) given privacy parameter $\epsilon, \delta$, total number of iteration $T$ and sampling rate $q = |\mathcal{B}|/N$, where $N$ is the total number of samples in the training set.

In the central setting, where a single trainer possesses all data, existing studies on fine-tuning LLM with DP guarantees mainly adopt DP-SGD as the optimization algorithm. Yu et al. (2021a) studied the effect of parameter-efficient algorithms in private fine-tuning. Li et al. (2021a; 2022) found that although the number of trainable parameters has been significantly reduced for PEFT, the performance of private fine-tuning is not significantly better, which might be contrary to traditional beliefs (Bassily et al., 2014).

**Different DP settings in FL.** Generally, there are two different levels of differential privacy protection in federated learning, depending on whether the federated aggregation server is trusted by the clients or not. The first setting assumes that the server is trusted, the model updates are shared to the server without privacy concerns; this privacy guarantee is on the final output model achieved by randomization in the global aggregated update on the server side (McMahan et al., 2017b). A stronger privacy setting is to forgo the trustworthy server assumption and ensure the shared update from each client is already differentially private (Li et al., 2021b; Wu et al., 2020; Qu et al., 2021).

In this paper, we adopt the *stronger privacy setting*, ensuring that any shared information (i.e., updates of model parameters) from local clients to server satisfies DP. By DP's properties, including parallel composition, sequential composition and resistance to post-processing (Dwork et al., 2006; Abadi et al., 2016; Li et al., 2021b), the final model automatically satisfies DP globally.

## 3 LoRA in privacy-preserving FL

In this paper, we focus on LoRA, one of the most promising PEFT methods in the central setting, LoRA has been shown to exhibit better performance than other PEFT methods in the federated setting Kuang et al. (2023). The core idea of LoRA is to constrain the weight update on the model by a low rank decomposition,

$$\boldsymbol{W}_0 + \Delta\boldsymbol{W} = \boldsymbol{W}_0 + \boldsymbol{B}\boldsymbol{A}. \qquad (1)$$

Instead of training the entire weight matrix $\boldsymbol{W}_0 \in \mathbb{R}^{d \times k}$ composing $\boldsymbol{\Phi}$, the updates are performed on $\boldsymbol{A} \in \mathbb{R}^{r \times k}$ and $\boldsymbol{B} \in \mathbb{R}^{d \times r}$ composing $\boldsymbol{\Theta}$. With $r << \min(d, k)$, the number of trainable parameters $|\boldsymbol{\Theta}|$ is reduced by an order of $O(r / \min(d, k))$ compared to full fine-tune with size $|\boldsymbol{\Phi}|$.

In order to recover the performance of raw model at the start of training, and keep the weights trainable through back-propagation, $\boldsymbol{A}$ uses random Gaussian initialization, while $\boldsymbol{B}$ is set to zero. The product matrix is additionally scaled by a factor $\alpha/r$. $\alpha$ also has influence on the performance of LoRA and is required to be tuned.

**Discordance 1: Data heterogeneity and model-averaging introduce interference to LoRA.** The performance of vanilla LoRA is negatively affected when faced with cross-silo FL tasks with data heterogeneity (Babakniya et al., 2023). Notice that the loss for back-propagation is computed on the composition of raw model parameters and the product of $\boldsymbol{A}$ and $\boldsymbol{B}$ (as Equation 1), and LoRA performs optimization over $\boldsymbol{A}$ and $\boldsymbol{B}$ jointly on client side. This implies that the problem is approximately optimized as a locally semi-quadratic problem (suppose the model is locally linear when learning rate is small). However, when the server performs aggregation on the server side, $\boldsymbol{A}$ and $\boldsymbol{B}$ are averaged separately following vanilla FedAvg McMahan et al. (2017a). The product of the averaged $\boldsymbol{A}$ and $\boldsymbol{B}$ involves additional terms that may neither benefit the optimization on the clients' loss or FL global loss.

For example, consider a FL task involving two clients with datasets of a same size. If clients locally fine-tune on full parameters and the server aggregates with FedAvg, the new model parameters can be represented as the following:

$$\boldsymbol{W}^+ = \frac{1}{2}(\boldsymbol{W}_1 + \boldsymbol{W}_2) = \boldsymbol{W}_0 + \frac{1}{2}(\Delta \boldsymbol{W}_1 + \Delta \boldsymbol{W}_2), \text{ where } \boldsymbol{W}_i = \boldsymbol{W}_0 + \Delta \boldsymbol{W}_i, i = 1, 2. \quad (2)$$

An implicit assumption ensuring the global convergence of FL algorithms is $\Delta \boldsymbol{W}_{global} \approx \frac{1}{2}(\Delta \boldsymbol{W}_1 + \Delta \boldsymbol{W}_2)$, where $\Delta \boldsymbol{W}_{global}$ is the update assuming the server can access all clients' dataset directly. When the clients use LoRA locally, we can also consider $\Delta \boldsymbol{W}_i \approx \boldsymbol{B}_i \boldsymbol{A}_i$. However, after using FedAvg to aggregate the trainable low-rank matrices, the server produces

$$\underbrace{\tilde{\boldsymbol{W}}^+ = \boldsymbol{W}_0 + \frac{1}{2}(\boldsymbol{B}_1 + \boldsymbol{B}_2) \times \frac{1}{2}(\boldsymbol{A}_1 + \boldsymbol{A}_2)}_{\text{Parameters after aggregation with LoRA + FedAvg}} \neq \underbrace{\boldsymbol{W}_0 + \frac{1}{2}(\boldsymbol{B}_1 \boldsymbol{A}_1 + \boldsymbol{B}_2 \boldsymbol{A}_2) = \boldsymbol{W}^+}_{\text{Ideal parameters following model-averaging}}. \quad (3)$$

Thus, it is possible for two clients in FL to converge to two different combinations of adaptation matrices $\boldsymbol{B}_i, \boldsymbol{A}_i$, yet when an aggregation such as FedAvg is applied in server, a linear combination does not necessarily provide good performance for the specific task. The difference between $\frac{1}{2}(\boldsymbol{B}_1 + \boldsymbol{B}_2) \times \frac{1}{2}(\boldsymbol{A}_1 + \boldsymbol{A}_2)$ and $\frac{1}{2}(\Delta \boldsymbol{W}_1 + \Delta \boldsymbol{W}_2)$ may become more significant when i) number of local update steps between aggregations is large and ii) the local datasets are different across clients.

This echoes with the "client-drift" phenomenon discussed by Karimireddy et al. (2020). "Client-drift" happens in heterogeneous FL when there is a difference between the average of local loss optima of clients and the optimum of the global loss, i.e. $\sum_i \boldsymbol{\Theta}_i^* \neq \boldsymbol{\Theta}_{global}^*$. It is caused by the local gradient dissimilarity among clients and slows down convergence. Since the parameters in LoRA are locally quadratic in construction, they are more prone to "client-drift" than a locally linear task such as full-fine-tuning.

**Discordance 2: The noise with DP-SGD can be amplified with LoRA.** Although LoRA and DP-SGD are the most popular methods in PEFT and privacy-preserved machine learning respectively, directly combining them together may not be the optimal choice. The discordance again comes from the semi-quadratic structure of LoRA. Consider the parameters after a single DP-SGD update. Even if no norm clipping operation is triggered, the parameters are updated as

$$\boldsymbol{W}_0 + (\boldsymbol{B} + \boldsymbol{\xi}_B)(\boldsymbol{A} + \boldsymbol{\xi}_A) = \boldsymbol{W}_0 + \boldsymbol{B}\boldsymbol{A} + \boldsymbol{\xi}_B \boldsymbol{A} + \boldsymbol{B}\boldsymbol{\xi}_A + \boldsymbol{\xi}_B \boldsymbol{\xi}_A,$$

where $\boldsymbol{\xi}_A$ and $\boldsymbol{\xi}_B$ consist of the Gaussian noises from DP-SGD. Three terms contain noise and the third term, $\boldsymbol{\xi}_B \boldsymbol{\xi}_A$, no longer follows a Gaussian distribution. This shows that noise is cascaded after the multiplication in LoRA, introducing additional difficulties for convergence in fine-tuning.

We provide synthetic verification with Figure 1. In this example, $\boldsymbol{W} \in \mathbb{R}^{1024 \times 1024}$ and rank $r = 8$. We plot the Frobenius norm of the noise matrices $\boldsymbol{\xi}_B \boldsymbol{\xi}_A$ and $\boldsymbol{\xi}_W$ for LoRA and full fine-tuning respectively. Due to the multiplication in LoRA by construction, the norm of noise scales quadratically with $\sigma$, and is significantly worse that full fine-tune when $\sigma$ exceeds 0.5.

For an FL algorithm with 1000 communication rounds, 10 local update steps and a dataset such as SST-2, using batch-size $\boldsymbol{B} = 200$, a DP guarantee with $\epsilon = 6, \delta = 1e - 5$ will require a noise factor of $\sigma = 0.99$. In this case, LoRA will produce approximately 3 times more noise compared to full model fine-tuning This could be an explanation to why LoRA does not significantly outperform full fine-tuning despite having less parameters, as reported in (Yu et al., 2021a; Li et al., 2022; Babakniya et al., 2023).

**Discordance 3: LoRA requires careful tuning on $\alpha$.**
In terms of the optimal scaling factor $\alpha$, empirical results (Kuang et al., 2023) have demonstrated that in many more complex tasks, a larger $\alpha$ shows higher performance after fine-tuning, yet as $\alpha$ increases, the algorithm becomes more and more unstable with much higher variance across different runs. According to Zhou & Cong (2017), the convergence speed for FedAvg-like algorithms is closely related to the objective function's smoothness factor $L$. As the scaling factor $\alpha$ increases, the problem becomes less smooth by construction, slowing down convergence.

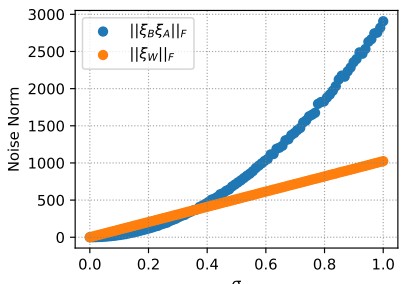

Figure 1: Frobenius norm of noise terms within a single update.

Furthermore, with increase of the scaling factor $\alpha$, the impact of noise on the model performance gets worse. This could be explained by the fact that as $\alpha$ increases, the update $\Delta \boldsymbol{A}$ becomes less significant compared to $\boldsymbol{A}_0$. Since $\boldsymbol{B}$ is initialized at 0, the gradient information in $\Delta \boldsymbol{B}$ becomes more important in comparison. Yet the gradient clipping and privacy engine sees the update on both $\boldsymbol{A}$ and $\boldsymbol{B}$ equally. Due to the imbalanced distribution of information in gradients, the algorithm suffers from either excessive information loss or excessive noise, a trade-off between increasing $\alpha$ for better performance and decreasing $\alpha$ to prevent noise-induced performance deterioration.

While searching for a good hyper-parameter $\alpha$ is important, hyper-parameter optimization (HPO) is usually costly (Khodak et al., 2021). Adding in $\alpha$ for HPO means extra communication and computation costs proportional to the search space size of the $\alpha$.

## 4 A SIMPLE RECEIPT: FFA-LoRA

In the previous section, we discussed the discordance between LoRA and privacy-preserved FL. Motivated by theory, we propose a simple modification to LoRA, Federated Freeze-A Low Rank Adapters, or FFA-LoRA for short. FFA-LoRA modifies the training process of LoRA by setting matrix $\boldsymbol{A}$ to fixed after initialization. That is, for a weight matrix $\boldsymbol{W} \in \mathbb{R}^{d \times k}$, we consider the model update to be projected to a low-rank matrix such that

$$\boldsymbol{W} = \boldsymbol{W}_0 + \Delta \boldsymbol{W} = \boldsymbol{W}_0 + \boldsymbol{B}\boldsymbol{A}_0, \text{ with } \boldsymbol{B} \in \mathbb{R}^{d \times r}, \boldsymbol{A}_0 \in \mathbb{R}^{r \times k}.$$

$\boldsymbol{W}_0$ is initialized as the pre-trained weight, and $\boldsymbol{A}_0$ follows a random Gaussian initialization. Following vanilla LoRA, we start with $\boldsymbol{B}_0 = 0$ so that the pre-trained model is recovered at the start of fine-tuning. The key difference is that we consider $\boldsymbol{B}$ trainable and keep both $\boldsymbol{W}_0$ and $\boldsymbol{A}_0$ frozen.

We note that our approach is somewhat similar to the works regarding the intrinsic dimension of deep models by (Li et al., 2018; Aghajanyan et al., 2020), however these works emphasis on the existence of low intrinsic dimensions in deep models and the generalization properties. We summarize the advantages of FFA-LoRA as the following.

**FFA-LoRA has no extra interference with data heterogeneity and model-averaging.** We reconsider the federated aggregation example in Section 3. In the heterogeneous setting, each client will generate a different $\Delta \boldsymbol{W}_i$. Since $\Delta \boldsymbol{W}_i \approx \boldsymbol{B}_i \boldsymbol{A}_0$ in FFA-LoRA, the update is more compatible with FedAvg and DP-SGD than LoRA. Similar to Equation 3, we write the the aggregation step of a two-client system for FFA-LoRA:

$$\tilde{\boldsymbol{W}}^+ = \boldsymbol{W}_0 + \frac{1}{2}(\boldsymbol{B}_1 + \boldsymbol{B}_2) \times \boldsymbol{A}_0 = \boldsymbol{W}_0 + \frac{1}{2}(\boldsymbol{B}_1 \boldsymbol{A}_0 + \boldsymbol{B}_2 \boldsymbol{A}_0) = \boldsymbol{W}^+. \tag{4}$$

Unlike LoRA in Equation 3, FFA-LoRA does not have the aggregation error term caused by low rank adaptation.

**FFA-LoRA works better with noise from DP-SGD.** Because FFA-LoRA no longer employs the locally semi-quadratic structure as LoRA, the noise in DP would not be amplified. When no norm clipping operation is triggered, the parameters are updated as $W_0 + (B + \xi_B)A_0 = W_0 + BA + \xi_B A$. This is because the trainable parameters are only the zero-initialized matrices $B$. The noise introduced by DP-SGD is only in the term $\xi_B A$ but without the $\xi_B \xi_B$ term, making FFA-LoRA less susceptible to noise than LoRA.

In addition, from an analytical perspective, if the model is Lipschitz smooth with respect to $W$, similar smoothness can be obtained for FFA-LoRA, but not for LoRA. We state our formal theorem and proof on the smoothness conditions of the two algorithm in the appendix. Convergence properties similar to (Zhou & Cong, 2017) can be derived from this theorem.

**FFA-LoRA does not rely on $\alpha$, and is equivalent to LoRA with $\alpha = \infty$.** In our previous discussion of LoRA's reliance on $\alpha$ in some tasks, this reliance on $\alpha$ is circumvented in FFA-LoRA. We can view the set of trainable parameters $\Theta$ as a dynamical system, a time-dependent series $\{\Theta_t\}_{t \in [T]}$ generated by the the FFA-LoRA algorithm. We present the following theorem to illustrate the connection between $\alpha$ and $\eta$ in FFA-LoRA.

**Theorem 1.** *For local updates with the same initial condition on* $W$, *vanilla LoRA update with scaling factor* $\alpha_{LoRA}$ *produces trajectory* $\{W^k_{\alpha_{LoRA}}\}_{k \in [K]}$, *and* FFA-LoRA *with scaling* $\alpha_{FFA}$ *produces trajectory* $\{W^k_{\alpha_{FFA}}\}_{k \in [K]}$. *Then we have*

$$\lim_{\alpha_{LoRA} \to \infty} W^k_{\alpha_{LoRA}} = W^k_{\alpha_{FFA}}, \text{ for all } k, \alpha_{FFA}. \tag{5}$$

We refer to the appendix for proof of Thm. 1. It is evident that introducing the scaling factor when fine-tuning with FFA-LoRA is unnecessary. However, the same does not apply to LoRA. For the case of LoRA, both $A$ and $B$ are trainable by construction, however, $A_0$ is initialized with Gaussian distribution, away from 0. For LoRA, when $\alpha$ is different, the initialization point is different, if we want two selection of $\alpha$ to have the same performance, we need to also change the variance of A's initialization.

For vanilla LoRA, as discussed in Section 3, as $\alpha$ increases and $\eta$ decreases, the update on $A$ become less significant compared to $A_0$. As $\alpha \to \infty$, there is almost zero change to be made on $A$, i.e. $A \sim A_0$. Yet the update on B is just as significant, making the dynamics of LoRA infinitely close to FFA-LoRA when $\alpha$ approaches infinity.

**FFA-LoRA saves computation and communication.** Since $A_0$ is fixed after initialization in FFA-LoRA, the total number of trainable parameters in the model is effectively halved compared to LoRA. This leads to the most straightforward advantage in the efficiency of computation and communication. Meanwhile, since we get same performance for a wide range of $\alpha$ in FFA-LoRA as long as the learning rate is scaled accordingly, we can fix $\alpha$ and only search for other hyper-parameters such as learning rate in HPO.

We note that subsequent to the submission of this paper, multiple new studies (Zhang et al., 2023; Zhu et al., 2024; Hao et al., 2024) have also considered similar approaches. While this paper distinctly considers the federated and privacy related properties, the succeeding papers can serve as verification of the effectiveness of FFA-LoRA. Another intuitive approach towards the problem at hand is to alternatively update the two LoRA weights. While this update method exhibit similar properties, it is empirically shown to be slow to converge.

In general, not only does FFA-LoRA provide higher efficiency compared to LoRA, FFA-LoRA is also able to preserve all the benefits of LoRA, while avoiding the shortcomings of LoRA as mentioned previously in Section 3.

## 5 EXPERIMENTS

In this section, we evaluate and compare the performance of FFA-LoRA with LoRA on two LMs, RoBERTa (Liu et al., 2019) and LLaMA (Touvron et al., 2023). We show that our approach consistently perform better for different types of tasks. We first evaluate the language understanding tasks from the GLUE benchmark(Wang et al., 2018) including MNLI, SST2, QNLI and QQP using the RoBERTa model. For language generation tasks, we use the LLaMA model with experiment settings provided by (Kuang et al., 2023) as benchmark and use the GSM-8K dataset for evaluation. All experiments were run using NVIDIA Tesla A100 GPUs with half-precision enabled for efficiency.

Our experiments are organized as follows: We provide the overall performance comparison of FFA-LoRA and LoRA in Section 5.1 (Table 1, 3). Questions regarding the critical factors of convergence are answered in Section 5.2 (Table 2, 4, 7). The evaluation on language generation tasks are provided in Section 5.3.

We note that our results do not exactly match the centralized PEFT results presented in (Hu et al., 2021) and (Yu et al., 2021a) due to the additional introduction of federated communication/aggregation and data heterogeneity in our setup. Our experiments with LoRA is able to match LoRA's performance reported in (Hu et al., 2021) in the centralized setting.

## 5.1 PERFORMANCE OF FFA-LoRA AND LoRA IN LANGUAGE UNDERSTANDING TASKS

Our experiments on language understanding tasks are based on RoBERTa-Large (355M) (Liu et al., 2019), a popular choice that has been widely adopted in many research studies for its robustness and versatility. We start from a pre-trained model available from the HuggingFace library.

All our experiments with LoRA and FFA-LoRA are run in a 3-client cross-silo federated setting. Data on clients are randomly split among all clients sampled to fit certain proportions to ensure strong data heterogeneity. For the heterogeneous setting, we split data based on their labels, we use $[0.1, 0.9], [0.9, 0.1], [0.5, 0.5]$ data split for binary classification tasks and $[0.9, 0.05, 0.05], [0.05, 0.9, 0.05], [0.05, 0.05, 0.9]$ for three-class classification tasks. In order to make a fair comparison, we keep the batch-size $B = 200$ and total communication round to 1000, the local update steps to 10, the same across all experiments. All experiments use the same SGD (DP-SGD for the experiments with privacy guarantees) optimizer, all the transformer-related hyper-parameters such as sequence length $l_{seq} = 128$, are kept to be consistent with previous studies (Hu et al., 2021). The classification head of the LM is frozen after initialization, and we add adapters to both the attention layers and the feed-forward layers and choose a scaling factor $\alpha = 8$ for LoRA. The same scaling factor $\alpha$ is applied to FFA-LoRA for the sake of consistency, although it is not needed as stated in Section 4.

**Experiments with differential privacy guarantees.** We report the best result from a set of experiments run with learning rate $\eta \in \{0.01, 0.02, 0.05, 0.1\}$ for LoRA and $\eta \in \{0.1, 0.2, 0.5, 1\}$ for FFA-LoRA. The batch-size and total number of update steps are kept to be the same across different tasks. We fix the rank $r = 8$ for both algorithms. In terms of privacy parameters, we use $\delta = 1e-5$ and three different choices of privacy budget $\epsilon \in \{6, 3, 1\}$. Given the sampling rate, total step number and privacy requirement $\epsilon, \delta$, we use the privacy accountant from Opacus (Yousefpour et al., 2021) to calculate the noise scale $\sigma$ for all our experiments. The optimal clipping threshold is determined from a grid search of $C \in \{2, 5, 10\}$. The results are presented in Table 1. To ensure the privacy guarantees have been met in our experiments, we refer to Section A.5 in the appendix for technical analysis.

The introduction of DP significantly degrades algorithm performance across every task for both FFA-LoRA and LoRA, yet FFA-LoRA offers better performance with and without privacy. We note that the biggest performance gap occurs in the MNLI task, which is a three-class classification task with the strongest level of data heterogeneity across agents. This performance gap demonstrates that FFA-LoRA is more suitable for tasks where heterogeneity is strong.

| Priv. Budget | Method | MNLI (matched) | MNLI (mismatched) | SST-2 | QQP | QNLI |
|---|---|---|---|---|---|---|
| Non Private | LoRA | $82.03_{\pm 10.7}$ | $82.50_{\pm 10.9}$ | $94.32_{\pm 2.1}$ | $83.51_{\pm 3.3}$ | $88.95_{\pm 6.7}$ |
| | FFA-LoRA | $85.05_{\pm 1.1}$ | $85.62_{\pm 1.0}$ | $94.32_{\pm 1.7}$ | $84.35_{\pm 0.6}$ | $90.35_{\pm 1.9}$ |
| $\epsilon = 6$ | LoRA | $39.46_{\pm 14.3}$ | $39.69_{\pm 14.8}$ | $93.70_{\pm 0.5}$ | $82.11_{\pm 1.0}$ | $84.99_{\pm 1.1}$ |
| | FFA-LoRA | $78.81_{\pm 0.8}$ | $80.00_{\pm 0.7}$ | $93.73_{\pm 0.3}$ | $83.31_{\pm 0.4}$ | $87.27_{\pm 1.0}$ |
| $\epsilon = 3$ | LoRA | $35.82_{\pm 8.9}$ | $35.85_{\pm 9.1}$ | $93.32_{\pm 0.5}$ | $82.08_{\pm 0.7}$ | $83.94_{\pm 0.6}$ |
| | FFA-LoRA | $77.42_{\pm 0.8}$ | $78.69_{\pm 0.8}$ | $93.59_{\pm 0.3}$ | $83.03_{\pm 0.4}$ | $86.18_{\pm 1.7}$ |
| $\epsilon = 1$ | LoRA | $33.80_{\pm 1.6}$ | $33.80_{\pm 1.5}$ | $92.14_{\pm 0.6}$ | $81.28_{\pm 0.7}$ | $78.93_{\pm 6.8}$ |
| | FFA-LoRA | $75.05_{\pm 1.3}$ | $76.50_{\pm 1.3}$ | $92.46_{\pm 0.5}$ | $82.50_{\pm 0.4}$ | $81.53_{\pm 1.4}$ |

Table 1: Experiments of FFA-LoRA and LoRA with differential privacy guarantees, accuracy (%) evaluated across 20 runs with mean and standard deviation.

## 5.2 ABLATION STUDY

Although FFA-LoRA is shown to be effective under federated settings and with private guarantees, previous works have also provided studies on the impact of the other hyper-parameters in LoRA algorithm. In order to provide a more comprehensive evaluation of the three discordances discussed in Section 3, we still need to answer the following questions:

- How does data heterogeneity affect performance of FFA-LoRA and LoRA?
- What is the impact of adapter parameter budget ($r$) for FFA-LoRA and the relationship between adapter parameter budget ($r$) and privacy budget ($\epsilon$) of DP-SGD?
- How do FFA-LoRA and LoRA behave when we choose different $\alpha$ for scaling?
- How does different initialization on $\boldsymbol{A}$ affect performance?

We answer the questions above with the following experiments.

**How does data heterogeneity affect performance of FFA-LoRA and LoRA?** Our discussion in Section 3 stated that LoRA is not compatible with FedAvg when there is strong heterogeneity among clients. For verification, we consider the four tasks with both homogeneous and heterogeneous data, and provide the experiment results below. The severe heterogeneity case corresponds to the data distribution provided in Section 5.1, while data is split with $[0.15, 0.85], [0.85, 0.15], [0.5, 0.5]$ and $[0.6, 0.2, 0.2], [0.2, 0.6, 0.2], [0.2, 0.2, 0.6]$ respectively in the mild heterogeneity configuration.

| Data Dist. | Method | MNLI (matched) | MNLI (mismatched) | SST2 | QQP | QNLI |
|---|---|---|---|---|---|---|
| i.i.d. | LoRA | 86.90 | 87.15 | 94.42 | 84.47 | 91.38 |
| | FFA-LoRA | 87.13 | 87.21 | 95.14 | 86.31 | 92.64 |
| mild het. | LoRA | 87.01 | 87.33 | 93.55 | 84.41 | 91.36 |
| | FFA-LoRA | 87.04 | 87.36 | 94.10 | 85.33 | 91.62 |
| severe het. | LoRA | 82.03 | 82.50 | 94.32 | 83.51 | 88.95 |
| | FFA-LoRA | 85.05 | 85.62 | 94.32 | 84.35 | 90.35 |

Table 2: Prediction accuracy (%) comparison between i.i.d. and non-i.i.d. data distribution.

It is evident that FFA-LoRA behaves better than LoRA in both i.i.d. and non-i.i.d. settings, but the performance is similar in the privacy-free setting.

**What is the impact of adapter parameter budget ($r$) for FFA-LoRA and the relationship between adapter parameter budget ($r$) and privacy budget ($\epsilon$) of DP-SGD?**

We first evaluate the performance of FFA-LoRA and LoRA without the consideration of privacy, we use the *mild heterogeneity* data distribution and keep the batch-size and total number of update steps to be the same across different tasks. We experiment with rank $r \in \{2, 4, 8, 16\}$ on four tasks, and report the best accuracy.

The results are shown in Table 3. From the *subspace similarity* discussions in LoRA, we note that increasing rank does not necessarily increase information from the gradients, similar observations can be found in our experiments. Based on the results, we can see that FFA-LoRA has better performance in the majority of tasks, regardless of the trainable parameter number. In fact, due to the reduction of trainable parameters in FFA-LoRA, we should compare between FFA-LoRA and LoRA with the same parameter budget (i.e. compare FFA-LoRA $r = 16$ with LoRA $r = 8$). In this case, the advantage of FFA-LoRA over LoRA becomes more apparent.

Although there have been multiple studies on the performance of LoRA with DP, the relationship between rank $r$ and privacy budget $\epsilon$ is unclear. We present the experiments below and compare the impact of rank $r$ on FFA-LoRA versus LoRA on the *QNLI* dataset. We use a privacy budget of $\epsilon \in \{6, 3, 1\}$ with rank $r \in \{2, 4, 8, 16\}$. The results are shown in Table 4.

In our experiments, we find that as the privacy requirements gets stronger, the performance difference of LoRA between different rank $r$ becomes more and more apparent, yet for FFA-LoRA, the algorithm is still able to output relatively stable performance on a wide range of rank selections.

**How do FFA-LoRA and LoRA behave when we choose different $\alpha$ for scaling?** As mentioned previously, LoRA requires a good scaling factor $\alpha$ in order to achieve a good performance. It has

| Method | # of params (million) | MNLI (matched) | MNLI (mismatched) | SST-2 | QQP | QNLI |
|---|---|---|---|---|---|---|
| LoRA (rank 16) | 3.15 (0.877%) | 87.43 | 87.47 | 93.98 | 84.79 | 91.92 |
| LoRA (rank 8) | 1.57 (0.440%) | 87.01 | 87.33 | 93.55 | 84.41 | 91.36 |
| LoRA (rank 4) | 0.79 (0.220%) | 86.07 | 86.41 | 93.89 | 83.71 | 91.51 |
| LoRA (rank 2) | 0.39 (0.110%) | 85.83 | 86.52 | 93.58 | 83.00 | 91.76 |
| FFA-LoRA (rank 16) | 1.57 (0.440%) | 85.82 | 86.38 | 95.30 | 84.89 | 91.65 |
| FFA-LoRA (rank 8) | 0.79 (0.220%) | 87.04 | 87.36 | 94.10 | 85.33 | 91.62 |
| FFA-LoRA (rank 4) | 0.39 (0.110%) | 85.61 | 86.11 | 94.47 | 84.64 | 91.38 |
| FFA-LoRA (rank 2) | 0.20 (0.055%) | 84.89 | 85.75 | 94.18 | 84.92 | 90.98 |

Table 3: Prediction accuracy (%) comparison on FFA-LoRA and LoRA with different ranks.

| privacy budget | Method | $r = 16$ | $r = 8$ | $r = 4$ | $r = 2$ |
|---|---|---|---|---|---|
| Non-Private | LoRA | 91.92 | 91.36 | 91.51 | 91.76 |
| | FFA-LoRA | 91.65 | 91.62 | 91.38 | 88.56 |
| $\epsilon = 6$ | LoRA | 86.87 | 86.45 | 85.24 | 83.54 |
| | FFA-LoRA | 87.33 | 87.57 | 86.74 | 86.31 |
| $\epsilon = 3$ | LoRA | 86.23 | 86.05 | 85.35 | 85.57 |
| | FFA-LoRA | 86.36 | 86.98 | 86.22 | 85.08 |
| $\epsilon = 1$ | LoRA | 80.54 | 81.45 | 58.30 | 58.15 |
| | FFA-LoRA | 81.87 | 83.01 | 82.06 | 82.64 |

Table 4: Prediction accuracy (%) of FFA-LoRA and LoRA across privacy and parameter budgets.

been shown in proof of Thm. 1 that the scaling factor does not affect the overall performance of the algorithm. We conducted experiments with a selection of different $\alpha$, and refer to A.6 in the appendix for the details and discussion.

**How does different initialization on $A$ affect performance?** Since our proposed FFA-LoRA sets $A$ as fixed throughout the fine-tuning process, a natural question would be regarding the initialization of $A$. We provide a discussion in Appendix A.8.

## 5.3 EXTENDING BEYOND LANGUAGE CLASSIFICATION

We next consider the task of Natural Language Generation (NLG) with LLaMA-7B, a more sophisticated model with significantly more parameters.

Our method has achieved an accuracy of $17.12\%$ on the task of GSM-8K, significantly better than the best performance of LoRA at $15.68\%$ ($15.31\%$ reported in (Kuang et al., 2023)). It is also the best results on fine-tuning LLaMA with GSM-8K to the best of our knowledge.

For an additional dataset on the computer vision task. We use the pre-trained vision transformer (Dosovitskiy et al., 2020) and consider the task of fine-tuning on the Food-101 dataset Bossard et al. (2014). In short, the algorithms performs similarly compared to the language classification tasks.

We report the details of the two experiments above in Appendix A.3 and A.7 respectively.

## 6 CONCLUSION

In this paper, we discussed how to improve LoRA in the context of privacy-preserving federated learning. An in-depth analysis was provided on LoRA's deficient performance in FL and with DP guarantees. We proposed a modification to LoRA named FFA-LoRA, which is theoretically motivated, empirically verified and computationally more efficient. Beyond the scope of this paper, FFA-LoRA could motivate more interesting problems related to PEFT for future study. For instance, we provide some preliminary results in Appendix A.4 to motivate future studies on algorithms that are even more parameter-efficient for federated LLM fine-tuning, one potential future direction is alternative initialization methods for matrices such as the orthogonal initialization. From a theoretical perspective, FFA-LoRA could be related to random kernel methods due to its pseudo-linear nature.

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

# A APPENDIX

## A.1 SMOOTHNESS ANALYSIS

**Theorem 2** (Smoothness conditions). *Assume that the loss function give weight and dataset is denoted $F(\mathbf{W}, D)$. For a low-rank decomposition on model parameter $\mathbf{W}$ such that $\mathbf{W}(\mathbf{A}, \mathbf{B}) = \mathbf{W}_0 + \mathbf{B}\mathbf{A}$ satisfying Equation (1). We have the following properties.*

1. *If $\mathbf{B}$ is trainable, $\mathbf{A}$ is fixed with $\|\mathbf{A}\| \leq C$ and $F(\mathbf{W}, D)$ is Lipschitz smooth with factor $L$. The loss function $F(\mathbf{W}(\mathbf{A}, \mathbf{B}))$ is Lipschitz smooth with respect to $\mathbf{B}$ with factor $LC^2$.*

2. *If both $\mathbf{A}$ and $\mathbf{B}$ are trainable and $F(\mathbf{W}, D)$ is Lipschitz smooth with factor $L$, the loss function $F(\mathbf{W}(\mathbf{A}, \mathbf{B}))$ has no Lipschitz smoothness guarantees.*

*All smoothness notions are defined with respect to matrix Frobenius norm, denoted as $\| \cdot \|$.*

*Proof.* First we show that, given $\mathbf{W}(\mathbf{A}, \mathbf{B}) = \mathbf{W}_0 + \mathbf{B}\mathbf{A}$, and the gradient on $\mathbf{W}$ is denoted as $\nabla_W F$, then we can write the gradients on matrix $\mathbf{B}$ as $\nabla_B F = \nabla_W F \mathbf{A}^T$, since

$$\langle \mathbf{B}_1 - \mathbf{B}_2, \nabla_B F \rangle = \langle \mathbf{W}(\mathbf{A}, \mathbf{B}_1) - \mathbf{W}(\mathbf{A}, \mathbf{B}_2), \nabla_W F \rangle$$
$$= \langle \mathbf{B}_1 \mathbf{A} - \mathbf{B}_2 \mathbf{A}, \nabla_W F \rangle$$
$$= \langle \mathbf{B}_1 - \mathbf{B}_2, \nabla_W F \mathbf{A}^T \rangle$$

Similarly, we have $\nabla_A F = \mathbf{B}^T \nabla_W F$. Using the gradients on $\mathbf{A}$ and $\mathbf{B}$, we provide the proof for all the properties.

1. For property 1, we know that for any given $\mathbf{B}_1, \mathbf{B}_2$,

$$\|\nabla_B F(\mathbf{W}(\mathbf{A}, \mathbf{B}_1)) - \nabla_B F(\mathbf{W}(\mathbf{A}, \mathbf{B}_2))\|$$
$$= \|\nabla_W F(\mathbf{W}(\mathbf{A}, \mathbf{B}_1)) \mathbf{A}^T - \nabla_W F(\mathbf{W}(\mathbf{A}, \mathbf{B}_2)) \mathbf{A}^T\|$$
$$\leq L \|\mathbf{W}(\mathbf{A}, \mathbf{B}_1) - \mathbf{W}(\mathbf{A}, \mathbf{B}_2)\| \|\mathbf{A}\|$$
$$\leq L \|\mathbf{B}_1 - \mathbf{B}_2\| \|\mathbf{A}\|^2$$
$$\leq LC^2 \|\mathbf{B}_1 - \mathbf{B}_2\|$$

2. For the second property, for the ease of notation, we introduce the stacked variable $\mathbf{x} := [\mathbf{A}, \mathbf{B}]$. We construct a counter-example such that the function is not Lipschitz smooth with respect to $\mathbf{x}$.

   We consider $\mathbf{W}, \mathbf{A}, \mathbf{B} \in \mathbb{R}^{d \times d}$, $F(\mathbf{W}) = \frac{1}{2}\|\mathbf{W}\|^2$ with $\mathbf{W}_0 = 0$. Then we consider a sequence $\{\mathbf{x}_k\}_{k \in \mathbb{N}}$ such that $\mathbf{x}_k = [\mathbf{A}_k, \mathbf{B}_k] = [k\mathbf{I}_d, k\mathbf{I}_d]$, then

$$\lim_{k \to \infty} \frac{\|\nabla_x \mathbf{W}(\mathbf{A}_k, \mathbf{B}_k) - \nabla_x \mathbf{W}(\mathbf{A}_0, \mathbf{B}_0)\|}{\|\mathbf{x_k} - \mathbf{x_0}\|}$$
$$= \lim_{k \to \infty} \frac{\|\nabla_A \mathbf{W}(\mathbf{A}_k, \mathbf{B}_k) - \nabla_A \mathbf{W}(\mathbf{A}_0, \mathbf{B}_0)\| + \|\nabla_B \mathbf{W}(\mathbf{A}_k, \mathbf{B}_k) - \nabla_B \mathbf{W}(\mathbf{A}_0, \mathbf{B}_0)\|}{\|\mathbf{A}_k - \mathbf{A}_0\| + \|\mathbf{B}_k - \mathbf{B}_0\|}$$
$$= \lim_{k \to \infty} \frac{\|k^3 \mathbf{I}_d\| + \|k^3 \mathbf{I}_d\|}{\|k \mathbf{I}_d\| + \|k \mathbf{I}_d\|}$$
$$= \infty$$

   From the existence of the above counter-example, we can see that although $F(\mathbf{W})$ is 1-Lipschitz smooth, the function is not smooth with respect to $\mathbf{x}$.

$\square$

## A.2 PROOF FOR THEOREM 1

*Proof.* The theorem starts with initial condition on $\mathbf{W}$, since $\mathbf{W} = \mathbf{W}_0 + \alpha \mathbf{B}_\alpha \mathbf{A}_\alpha$ and that the initialization of $\mathbf{A}$ is non-zero, this condition implies that $\mathbf{A}_\alpha = \mathbf{A}_1 = \mathbf{A}$, and $\mathbf{B}_\alpha = \frac{1}{\alpha} \mathbf{B}_1$. Now we compare the update of the two algorithms given the same initial conditions.

From Theorem 1 we know that for FFA-LoRA, different $\alpha_{FFA}$ does not affect its dynamics, without loss of generality, we consider the case where $\alpha_{FFA} = 1$.

The FFA-LoRA update is as follows, the only update is on $B$:

$$W_{FFA}^{k+1} = W_0 + 1 \times B^{k+1} A^k = W_0 + (B^k - \eta \nabla B^k) A^k = W^k - \eta \nabla B^k A^k$$

The rest of the proof is given by induction, as long as the limit holds for the $k+1$-th local iteration given that the $k$-th iteration holds.

Without the loss of generality, we first consider when $\alpha_{LoRA} = 1$, then for iteration $k$, we denote the learning rate as $\eta_1$, denote the matrices and their gradient as $\mathbf{A}_1^k, \mathbf{B}_1^k$ and $\nabla \mathbf{A}_1^k, \nabla \mathbf{B}_1^k$, respectively. And by definition, we have the update that

$$\mathbf{A}_1^{k+1} \leftarrow \mathbf{A}_1^k - \eta_1 \nabla \mathbf{A}_1^k$$
$$\mathbf{B}_1^{k+1} \leftarrow \mathbf{B}_1^k - \eta_1 \nabla \mathbf{B}_1^k$$

And the update of the original weight matrix $W$ becomes

$$W_1^{k+1} = W_0 + \Delta W_1^{k+1} = W_0 + (\mathbf{B}_1^k - \eta_1 \nabla \mathbf{B}_1^k)(\mathbf{A}_1^k - \eta_1 \nabla \mathbf{A}_1^k)$$
$$= W_1^k - \eta_1 \left( \nabla \mathbf{B}_1^k \mathbf{A}_1^k + \mathbf{B}_1^k \nabla \mathbf{A}_1^k \right) + \eta^2 \nabla \mathbf{B}_1^k \nabla \mathbf{A}_1^k$$

Since LoRA do not satisfy the conditions provided in Theorem 1, changing $\alpha_{LoRA}$ will affect its updates. When we choose a different $\alpha_{LoRA} = \alpha$ and corresponding $\eta_\alpha = \frac{\eta_1}{\alpha^2}$, we can write the update of LoRA as

$$\mathbf{A}_\alpha^{k+1} \leftarrow \mathbf{A}_\alpha^k - \eta_\alpha \nabla \mathbf{A}_\alpha^k = \mathbf{A}_\alpha^k - \frac{\eta_1}{\alpha} \nabla \mathbf{A}_\alpha^k = \mathbf{A}_1^k - \frac{\eta_1}{\alpha} \nabla \mathbf{A}_1^k$$

$$\mathbf{B}_\alpha^{k+1} \leftarrow \mathbf{B}_\alpha^k - \eta_\alpha \nabla \mathbf{B}_\alpha^k = \mathbf{B}_\alpha^k - \frac{\eta_1}{\alpha} \nabla \mathbf{B}_\alpha^k = \frac{1}{\alpha} \mathbf{B}_1^k - \frac{\eta_1}{\alpha} \nabla \mathbf{B}_1^k$$

$$W_\alpha^{k+1} = W_0 + \alpha(\frac{1}{\alpha} \mathbf{B}_1^k - \frac{\eta_1}{\alpha} \nabla \mathbf{B}_1^k)(\mathbf{A}_1^k - \frac{\eta_1}{\alpha} \nabla \mathbf{A}_1^k)$$

$$= W_0 + \mathbf{B}_1^k \mathbf{A}_1^k - \eta_1 \nabla \mathbf{B}_1^k \mathbf{A}_1^k - \frac{\eta_1}{\alpha} \mathbf{B}_1^k \nabla \mathbf{A}_1^k - \frac{\eta_1^2}{\alpha} \nabla \mathbf{B}_1^k \nabla \mathbf{A}_1^k$$

Therefore we have

$$\lim_{\alpha_{LoRA} \to \infty} W_{\alpha_{LoRA}}^{k+1} = \lim_{\alpha \to \infty} W_0 + \mathbf{B}_1^k \mathbf{A}_1^k - \eta_1 \nabla \mathbf{B}_1^k \mathbf{A}_1^k - \frac{\eta_1}{\alpha} \mathbf{B}_1^k \nabla \mathbf{A}_1^k - \frac{\eta_1^2}{\alpha} \nabla \mathbf{B}_1^k \nabla \mathbf{A}_1^k$$
$$= W_1^k - \eta_1 \nabla \mathbf{B}_1^k \mathbf{A}_1^k$$
$$= W_{FFA}^{k+1}$$

Which completes our proof.

$\square$

### A.3 LLaMA Experiments and Details

Similar to RoBERTa, LLaMA is also widely used and offers competitive results for its network size. We evaluate both LoRA and FFA-LoRA with the GSM-8K dataset using the same set of hyper-parameters listed by (Kuang et al., 2023). Our method has achieved an accuracy of $17.12\%$ on the task of GSM-8K, significantly better than the best performance of LoRA at $15.68\%$ ($15.31\%$ reported in (Kuang et al., 2023)). It is also the best results on fine-tuning LLaMA with GSM-8K to the best of our knowledge. We sample some generated answers of LoRA and FFA-LoRA in Table 5. The answers generated by FFA-LoRA demonstrate better performance on longer questions with more complex contexts.

### A.4 A motivation for further reducing trainable parameters

Our approach named FFA-LoRA in Section 4 exhibits a number of theoretical benefits compared to LoRA, additionally, it also performs better and is more consistent as shown in Section 5. We can conclude that for PEFT with adapters, by freezing randomly initialized parameters and only train on

| Question: | A new program had 60 downloads in the first month. The number of downloads in the second month was three times as many as the downloads in the first month, but then reduced by 30% in the third month. How many downloads did the program have total over the three months? |
|---|---|
| FFA-LoRA: | The number of downloads in the first month was 60. The number of downloads in the second month was 3 times as many as the first month, so that is 3 * 60 = 180. The number of downloads in the third month was 180 - 30% = 180 - 54 = 126. The total number of downloads over the three months is 60 + 180 + 126 = 366. The answer is 366. |
| LoRA: | The number of downloads in the first month was 60. The number of downloads in the second month was 60 x 3 = 180. The number of downloads in the third month was 180 - 30 = 150. The answer is 150. |
| Question: | John decides to get a loan by mortgaging his home. His house is worth $250,000. He gets a loan worth 40% of that. He uses 60% of that to pay off his debts. How much money did he have leftover after paying debt? |
| FFA-LoRA: | John's house is worth 250,000 dollars. He gets a loan worth 250,000 x 40% = 100,000 dollars. He uses 100,000 x 60% = 60,000 dollars to pay off his debts. 100,000 - 60,000 = 40,000 dollars. The answer is 40,000. |
| LoRA: | John's house is worth 250,000 dollars. 40% of that is 250,000 x 0.4 = 100,000 dollars. 100,000 dollars is the amount of the loan. 100,000 / 0.6 = 166,666 dollars. 166,666 - 100,000 = 66,666 dollars. 66,666 dollars is the amount of money he has left. |
| Question: | Octavia wants to make a quilt for her sister. She is going to build it from small, medium, and large t-shirts that her family is done with. A small t-shirt is 3 square feet of fabric. A medium one is 4 square feet. A large one is 6 square feet. If she uses 11 small shirts, 8 medium, and 6 large shirts, how many square feet is the quilt? |
| FFA-LoRA: | A small shirt is 3 square feet. A medium shirt is 4 square feet. A large shirt is 6 square feet. 11 small shirts is 11 x 3 = 33 square feet. 8 medium shirts is 8 x 4 = 32 square feet. 6 large shirts is 6 x 6 = 36 square feet. 33 + 32 + 36 = 101 square feet. The answer is 101. |
| LoRA: | Octavia is going to use 11 small shirts, 8 medium, and 6 large shirts. 11 + 8 + 6 = 25. 3 square feet is 12 square feet. 25 * 12 = 300 square feet. The answer is 300 square feet. |

Table 5: Example of generated answer for GSM8K datasets

| Method | # of params | acc w/o DP | acc@$\epsilon = 6$ | acc@$\epsilon = 3$ |
|---|---|---|---|---|
| LoRA (rank 16) | 3145728 (0.877%) | 92.49% | 86.87% | 86.23% |
| LoRA (rank 4) | 786432 (0.220%) | 91.40% | 85.2% | 85.35% |
| FFA-LoRA (rank 16) | 1572864 (0.440%) | 92.49% | 87.33 % | 86.36% |
| FFA-LoRA (rank 4) | 393216 (0.110%) | 92.20% | 86.75% | 86.22% |
| QVP (rank 128) | 1572864 (0.412%) | 90.46% | 84.23% | 83.16% |
| QVP (rank 64) | 393216 (0.107%) | 90.17% | 86.41% | 84.44% |
| QVP (rank 32) | 98304 (0.0272%) | 87.31% | 85.69% | 84.31% |
| QVP (rank 16) | 24576 (0.00685%) | 83.40% | 84.44% | 83.67% |

Table 6: Comparison between LoRA, FFA-LoRA and QVP adapters, including number of trainable parameters.

the set of parameters that were initialized at $0$ is a valid and practical approach. This guided us to get an even more aggressive construction of adapters, which we refer to as QVP Adapters, formulated as below.

For a weight matrix $W \in \mathbb{R}^{d \times k}$, we consider the model update to be projected to a low-rank matrix such that

$$W = W_0 + \Delta W = W_0 + Q_0 V P_0,$$

where $Q_0 \in \mathbb{R}^{d \times r}, V \in \mathbb{R}^{r \times r}, P_0 \in \mathbb{R}^{r \times k}$. Similar to FFA-LoRA, $W_0$ is the pre-trained weight, and $P_0, Q_0$ follows a random Gaussian initialization. We consider $V$ trainable and start with $V_0 = 0, W_0, Q_0, P_0$ are kept frozen throughout the training process. We provide the performance of QVP adapters below in Table 6, and compare with LoRA and FFA-LoRA.

For the experiments where these algorithms have the same parameter budget ($r = 64$ for QVP versus $r = 4$ for FFA-LoRA, etc.), QVP do not perform as good as the previously mentioned algorithms. But a unique advantage offered by QVP adapters is that it is possible to even further reduce the number of trainable parameters, and the algorithm is still able to learn meaningful features from data. The same is impossible for LoRA and FFA-LoRA since the rank $r$ can not be smaller than 1 for these methods. Therefore, QVP is potentially useful in the case where the parameter budget is extremely constrained, such as local private training in mobile devices.

## A.5 DIFFERENTIAL PRIVACY GUARANTEE

We present the following corollary regarding the privacy guarantees in our experiments.

**Corollary 2.1** (Privacy Guarantee). *Given Theorem 1 with moments accountant in (Abadi et al., 2016), the parallel composition and resistance to post-processing of DP, the mechanism updating FFS-LoRA with locally ran DP-SGD and FedAvg can satisfy $(\epsilon, \delta)$-DP given $\forall i, q = \frac{|B_i|}{|N_i|}$, the number of total local updates $T$ of each client and $\sigma = O(\frac{q\sqrt{T \log(1/\delta)}}{\epsilon})$. (The exact $\sigma$ is computed by the Pytorch's Opacus package (Yousefpour et al., 2021) numerically given $q, T, \epsilon, \delta$).*

*Proof.* Firstly, we consider the local datasets $\{D_i\}_{i \in [n]}$ for the FL network to be disjoint chunks of the global dataset. The DP-SGD with FedAvg used in our paper to train LoRA or FFA-LoRA can be considered as (A) locally updating trainable parameters with DP-SGD, (B) averaging the trainable parameters from clients on the server, and (C) repeating the above two steps for some iterations. The privacy loss of (A) can be composed by moment accountants used in (Abadi et al., 2016). The privacy loss of all clients performing local updates can be composed by the parallel composition property of DP. The averaging on the server in (B) is a post-processing operation that does not introduce privacy loss. Privacy loss of multiple FL rounds of (C) can again be composed with moment accountants used in (Abadi et al., 2016). Eventually, we can convert the moment accountants to $(\epsilon, \delta)$-DP as Theorem 1 in (Abadi et al., 2016). □

## A.6 EXPERIMENTS WITH DIFFERENT SCALING FACTOR $\alpha$

We conducted experiments with a selection of different $\alpha$, we use $\alpha = 8, r = 8$ as baseline, and choose learning rate $\eta$ according to the learning rate scaling discussed above. Our results are shown

in Table 7. For FFA-LoRA, the performance using $\eta$ that scales with $\alpha$ is consistent across the wide range of $\alpha$. However for LoRA, the same relationship does not hold, and the performance of LoRA degrades drastically when $\alpha$ changes. Additional grid-search shows that LoRA still is able to converge with high accuracy with adequate learning rate, but finding an optimal learning rate given $\alpha$ is in general arduous. We note that the optimal $\eta$ for $\alpha = 256$ is the same for both FFA-LoRA and LoRA, consistent with our discussion in Section 4.

| Method | $\alpha = 2$ | $\alpha = 8$ | $\alpha = 16$ | $\alpha = 64$ | $\alpha = 256$ |
|---|---|---|---|---|---|
| LoRA (best LR) | 91.78% | 91.36% | 92.11% | 91.50% | 91.23% |
| LoRA (LR scaling) | 71.88% | 91.36% | 92.11% | 50.96% | 49.46% |
| FFA-LoRA (LR scaling) | 91.31% | 91.62% | 91.9% | 91.17% | 92.46% |

Table 7: Experiment with different scaling factor $\alpha$.

## A.7 COMPUTER VISION EXPERIMENTS

For context, we provide performance reported on huggingface as baseline. A centralized, fine-tuned model has an accuracy of 0.8539.

We first report the results in our centralized experimental setting in the table below. In this case there is no significant performance discrepancy between the two methods, implying that FFA-LoRA and vanilla has similar performance without consideration of DP and FL. This also aligns with our observations in previous experiments.

In terms of the federated case, we first report the iid setting. It can be seen that compared to LoRA, FFA-LoRA has both (a) better convergence and (b) less fluctuations in training. The findings align with our findings in language-related tasks, showing that the properties of LoRA being discussed in our paper are not limited to language tasks only.

| Method | Baseline | Cen. LoRA | Cen. FFA-LoRA | FL iid LoRA | FL non-iid FFA-LoRA |
|---|---|---|---|---|---|
| Accuracy | 85.39% | 86.18% | 85.83% | 81.33% | 82.10% |

Table 8: Performance of FFA-LoRA in vision transformer evaluated on Food-101 dataset.

## A.8 DIFFERENT MATRIX INITIALIZATION FOR A

Since our proposed FFA-LoRA sets $A$ as fixed throughout the fine-tuning process, a natural question would be regarding the initialization of $A$. We know that for a zero-initialized A matrix, neither LoRA nor FFA-LoRA are able to train any meaningful results. However, suppose that we have A to be full rank (which is also satisfied for any random initialization in general), there are a number of different initialization that we could utilize.

In the majority of this paper, we consider the same initialization as LoRA. Apart from Kaiming initialization, we consider orthogonal random initialization and using the top $r$ singular vectors of $W_0$ as matrix A. We provide some initial results in Table 9, it can be seen from the plot that matrix with orthogonal initialization seems to perform slightly better than the existing approach. However, the performance gap is not significant enough for a definitive answer.

| Method | QNLI mean | QNLI variance |
|---|---|---|
| Kaiming Init. | 91.84% | 0.38% |
| Orthogonal Init. | 92.16% | 0.83% |
| SVD Init. | 91.50% | 0.59% |

Table 9: Performance of algorithm under similar conditions and different initialization on matrix $A$.

