# OpenReview forum: "Improving LoRA in Privacy-preserving Federated Learning"
_ICLR.cc/2024/Conference — ICLR 2024 poster_

### Official Review · Reviewer_8DEX · 2023-10-29

**Soundness:** 3 good
**Presentation:** 3 good
**Contribution:** 3 good
**Rating:** 6
**Confidence:** 4

**Summary:**

The Federated Generative Learning (FGL) framework offers a novel approach to federated learning, leveraging foundational generative models like Stable Diffusion to generate training data from prompts shared by clients. Clients contribute class-level or instance-level prompts, encapsulating key features of their local data. The server, in turn, amalgamates these prompts and synthesizes corresponding training data for global model training. This approach trims down communication costs since only concise prompts, and not bulky gradients or models, are transferred. This system also boasts robustness to data diversity and has demonstrated superior performance – with just one communication round, it outdid FedAvg's 200 rounds in accuracy. When trialed on skewed ImageNet100 distributions, FGL exceeded FedAvg's performance by 30% in just five communication rounds. Apart from being efficient, FGL also enhances privacy, as prompts reveal lesser private data than traditional methods. Evaluations confirmed no private data memorization in the synthetic images and an enhanced resilience against membership inference attacks. However, challenges persist with non-IID data, intricate domains, and the potential risks associated with prompts.

**Strengths:**

1.	Clearly identifies limitations of vanilla LoRA in federated learning settings and provides theoretical analysis on the causes.
2.	Provides extensive experiments that demonstrate consistent improvements of FFA-LoRA over LoRA on multiple models, datasets, and conditions.
3.	Reduces communication costs and removes reliance on scaling hyperparameters compared to LoRA.

**Weaknesses:**

1.	Unclear how the approach performs under other challenges like adversarial attacks, concept drift, and personalization.
2.	The paper only evaluates NLP tasks with text data. Unclear if the benefits of FFA-LoRA generalize to other data types like image, speech, etc.
3.	The theoretical analysis and intuitions provided are informal. No formal convergence or privacy proofs given.

**Questions:**

please refer to the weakness

---

> ### Author Response · Authors · 2023-11-19
> **Response to Reviewer 8DEX**
>
> We thank the reviewer for the time and support of our paper as well as the valuable suggestions. Please see our response below with respect to the specific comments.
>
> > Unclear how the approach performs under adversarial attacks, concept drift, and personalization.
>
> The main objective of our paper is on the challenges when we consider the task of fine-tuning of LLMs, and impact of . Although these questions are important in the context of Fed Fine tuning LLMs. They are beyond the scope of this paper.
>
> > The paper only evaluates NLP tasks with text data. Unclear if the benefits of FFA-LoRA generalize to other data types like image, speech, etc.
>
> It would be interesting to see the performance of such fine-tuning in the context of computer vision and speech. We running experiments on these tasks and will update the results here once they are ready. We note that fine-tuning LLMs with DP is computationally intensive, and we need more time to finish the experiments.
>
> > The theoretical analysis and intuitions provided are informal. No formal convergence or privacy proofs given.
>
> Regarding the **scaling factor** $\alpha$ for FFA-LoRA and LoRA, we have updated our draft to include more formal statements, and provide an additional theorem to formally show that FFA-LoRA is in fact equivalent to LoRA with $\alpha \rightarrow \infty$.
> >> **Theorem 1 (Scaling Factor):** For local updates with the same initial condition on $\mathbf{W}$, vanilla LoRA update with scaling factor $\alpha\_{LoRA}$ produces trajectory $ \\{ W\_{\alpha\_{LoRA}}^k \\}\_{k\in [K]}$, and FFA-LoRA with scaling $\alpha\_{FFA}$ produces trajectory $\\{W\_{\alpha\_{FFA}}^k\\}\_{k\in [K]}$. Then we have
> >> $$\lim_{\alpha_{LoRA} \rightarrow \infty} W_{\alpha_{LoRA}}^k = W_{\alpha_{FFA}}^k, \text{ for all } k, \alpha_{FFA}.$$
>
> Regarding the **convergence study** of our proposed method. We first provide the following statements for some additional context on the theoretical aspects behind PEFT on LLMS.
> >> **Theorem 2 (Smoothness conditions):** Assume that the loss function given weight and dataset is denoted $F(\mathbf{W}, D)$. For a low-rank decomposition on model parameter $\mathbf{W}$ such that $\mathbf{W}(\mathbf{A}, \mathbf{B}) = \mathbf{W}\_0 + \mathbf{B} \mathbf{A}$ satisfying Equation (1). We have the following properties.
> >> 1. If $\mathbf{B} $ is trainable, $\mathbf{A}$ is fixed with $||\mathbf{A}|| \leq C$ and $F(\mathbf{W}, D)$ is Lipschitz smooth with respect to $\mathbf{W}$ with factor $L$. The loss function $F(\mathbf{W}(\mathbf{A}, \mathbf{B}))$ is Lipschitz smooth with respect to $\mathbf{B}$ with factor $LC^2$.
> >> 2. If both $\mathbf{A}$ and $\mathbf{B}$ are trainable and $F(\mathbf{W}, D)$ is Lipschitz smooth with respect to $\mathbf{W}$ with factor $L$, the loss function $F(\mathbf{W}(\mathbf{A}, \mathbf{B}))$ has no Lipschitz smoothness guarantees with respect to $\[\mathbf{A}, \mathbf{B}\]$.
> >>
> >> All the smoothness notions are defined with respect to the matrix Frobenius norm, which is denoted as $||\cdot||$.
>
> These results show that if the objective function loss defined on full fine-tuning with $\mathbf{W}$ satisfies technical conditions such as Lipschitz smoothness, the same assumptions would also hold for FFA-LoRA, but not for Vanilla LoRA.
>
> Suppose that the objective function $F$ satisfies the conditions presented in existing works to ensure convergence of full federated fine-tuning [1] or DP tuning [2] on $\mathbf{W}$, then we can get similar **convergence results for FFA-LoRA using the theorem above**.
>
> Regarding **differential privacy**, we formally state our corollary for privacy guarantee. Our analysis below is based on Theorem 1 of [3] and the parallel composition and resistance to post-processing of DP.
> >>**Corollary 3 (Privacy Guarantee):** Given Theorem 1 with moments accountant in [3],  the parallel composition and resistance to post-processing of DP, the mechanism updating FFS-LoRA with locally ran DP-SGD and FedAvg can satisfy $(\epsilon, \delta)$-DP given $\forall i, q=\frac{|B_i|}{|N_i|}$, the number of total local updates $T$ of each client and  $\sigma = O(\frac{q\sqrt{T\log(1/\delta)}}{\epsilon})$.
>
> (The exact $\sigma$ is computed by the Pytorch's Opacus package [4] numerically given $q, T, \epsilon, \delta$).
>
> We refer to our "**Response to All**" for detailed proof.
>
> ---
> [1] *Lian, Xiangru, Yijun Huang, Yuncheng Li, and Ji Liu.* "Asynchronous parallel stochastic gradient for nonconvex optimization." *Advances in neural information processing systems 28 (2015).*
>
> [2] *Wang, Di, Changyou Chen, and Jinhui Xu.* "Differentially private empirical risk minimization with non-convex loss functions." *In International Conference on Machine Learning, 2019.*
>
> [3] *Martin Abadi, Andy Chu, Ian Goodfellow, H Brendan McMahan, Ilya Mironov, Kunal Talwar, and
> Li Zhang.* Deep learning with differential privacy. *2016 ACM SIGSAC
> conference on computer and communications security*
>
> [4] <https://github.com/pytorch/opacus>

---

> > ### Comment · Reviewer_8DEX · 2023-11-22
> > **Thanks**
> >
> > Thanks for the authors' detailed explanations, most of my concerns have been addressed. Do you have to updated results on other datasets?

---

> ### Author Response · Authors · 2023-11-23
> **Reply to Comment**
>
> We thank the reviewer for their kind acknowledgment of our response and that our response addresses most of their concerns.
>
> We here provide an additional dataset on the computer vision task.
> We use the pre-trained vision transformer (https://huggingface.co/google/vit-base-patch16-224-in21k), and consider the task of fine-tuning on the Food-101 dataset (as shown on https://huggingface.co/datasets/food101) for image classification.
>
> For context, we provide performance reported on huggingface as baseline. A centralized, fine-tuned model has an accuracy of 0.8539.
>
> We first report the results in our centralized experimental setting in the table below.In this case there is no significant performance discrepancy between the two methods, implying that FFA-LoRA and vanilla has similar performance without consideration of DP and FL. This also aligns with Theorem 1 in Author's response.
>
> In terms of the federated case, we first report the iid setting. In this setting, under similar hyper-parameters, we provide the learning curve of both FFA-LoRA and LoRA (https://imgur.com/5xnXWSu). as well as a last k-iterated (k=50) averaged accuracy.
>
> It can be seen that compared to LoRA, FFA-LoRA has both (a) better convergence and (b) less fluctuations in training. The findings align with our findings in language-related tasks, showing that the properties of LoRA being discussed in our paper are not limited to language tasks only.
>
> The non-iid setting as well as the DP setting are more time-consuming, and we are still running more experiments. Due to the time limit, we will add them to the final version of our paper if the paper is accepted.
>
>
> | Method          | Accuracy |
> |-----------------|----------|
> | Baseline        | 0.8539   |
> | Centralized LoRA | 0.8618   |
> | Centralized FFA-LoRA | 0.8583   |
> | FL iid LoRA | 0.8133   |
> | FL iid FFA-LoRA | 0.8210   |

---

### Official Review · Reviewer_E2bX · 2023-10-31

**Soundness:** 2 fair
**Presentation:** 2 fair
**Contribution:** 2 fair
**Rating:** 5
**Confidence:** 3

**Summary:**

This paper presented an approach called Federated Freeze A LoRA (FFA-LoRA) to address the limitations of the low-rank adaptation method in federated learning setting. The limitations of the vanila low-rank adaptation include: 1) data heterogeneity, 2) amplication of difficiential privacy noise and 3) sensitivity to hyper-parameters. Authors provide empirical results, showing that the FFA-LoRA outperforms vanilla LoRA in federated learning settings.

**Strengths:**

1. The study on federated LoRA is timely.
2. The proposed approach is simple to implement.
3. The authors provide case studies to highlight the limitations of the vanilla LoRA and motivate their approach.

**Weaknesses:**

1. The benefit of FFA-LoRA on differential privacy (DP) is not very well backed by empirical evaluation. The performance gap between the vanilla LoRA and the proposed FFA-LoRA remains the same across various privacy budgets $\epsilon$, including $\epsilon = 0$. Such an empirical result suggests that the impact of DP noise is the same on both the vanilla LoRA and the proposed FFA-LoRA.

2. I do not see why the proposed FFA-LoRA is free from tuning the hyper-parameter $\alpha$. In Section 4, the authors claim that "FFA-LoRA does not rely on $\alpha$, and is equivalent to LoRA with $\alpha = \infty$". Such a claim, in fact, suggests that the $\alpha$ is fixed in FFA-LoRA. Then, in Theorem 1, the theoretical result suggests that tuning $\alpha$ is equivalent to tuning the learning rate $\eta$. I'm not able to fully follow the discussion here.

**Questions:**

1. Is the FFA-LoRA approach more sensitive to random initialization? Suppose a bad initialization sets $\mathbf{A} = \mathbf{0}$; is the model still trainable?

2. What's the variance in the experiment?

---

> ### Author Response · Authors · 2023-11-19
> **Response to Reviewer E2bX (Part 1)**
>
> > The benefit of FFA-LoRA on differential privacy (DP) is not very well backed by empirical evaluation. The performance gap between the vanilla LoRA and the proposed FFA-LoRA remains the same across various privacy budgets
> , including
> . Such an empirical result suggests that the impact of DP noise is the same on both the vanilla LoRA and the proposed FFA-LoRA.
>
> We note that the exact performance of the algorithms are In Table 1, although there are counter-examples, the performance gap between FFA and vanilla is still evident, with the largest gap occurring $\epsilon = 1$ when taking average over the 4 tasks. Therefore we can say that FFA is more suitable for higher noise.
> The individual performance of one task, $SST-2$ in this case, depends on many factors such as the difficulty of the task.
>
> We further point out that our claim on noise amplification is more apparent in Table 4. Where for rank as low as 2 or 4, Lora can not converge to any meaningful results when $\epsilon = 1$.
>
> Additionally, when noise is small (no significant noise amplifying), the impact of noise is not as big, as shown in Figure 1 of our paper. The simulation in Figure 1 also aligns with our observation in Table 1 and 4.
>
> Additionally, existing literature [1] (Figure 4) has shown that the relationship between performance gap and $\epsilon$ is not always significant, and varies between tasks. This observed behavior could be due to robustness of DPSGD-style algorithms. Currently, the dynamics of DPSGD optimization in deep neural networks is still not fully understood.
>
> > I do not see why the proposed FFA-LoRA is free from tuning the hyper-parameter
> . In Section 4, the authors claim that "FFA-LoRA does not rely on
> , and is equivalent to LoRA with
> ". Such a claim, in fact, suggests that the
>  is fixed in FFA-LoRA. Then, in Theorem 1, the theoretical result suggests that tuning
>  is equivalent to tuning the learning rate
> . I'm not able to fully follow the discussion here.
>
> We would like to first reiterate that for the proposed FFA-LoRA algorithm, as confirmed by Theorem 1 in our paper, $\alpha_{FFA}$ can be seen as fixed. However, this does not make FFA-LoRA equivalent to vanilla LoRA with some given $\alpha_{LoRA}$. As we have stated, FFA-LoRA with any $\alpha$ in fact has the same dynamic as LoRA with $\alpha_{LoRA} \rightarrow \infty$, we provide the details below:
>
> In order to reduce confusion and provide additional theoretical robustness to our paper, we provide an additional theorem as a more formal statement on the relationship between FFA-LoRA and vanilla LoRA. We refer to our "**Response to All**" for a more detailed explanation.
>
> > Is the FFA-LoRA approach more sensitive to random initialization? Suppose a bad initialization sets is the model still trainable?
>
> This is a really good question. In our experiments, all the $A_0$ matrices were randomly initialized under random Kaiming distribution ( a default initialization method is deep learning and is consistent with previous works in LoRA) and according to Theorem 1, the scaling on the $A_0$ is not important for FFA-LoRA.
> And we briefly discuss the impact of other types of initialization below.
>
> We know that for a zero-initialized A matrix, neither LoRA nor FFA-LoRA are able to train any meaningful results. However, suppose that we have A to be full rank (which is also satisfied for any random initialization in general), there are a number of different initializations that we could utilize, such as orthogonal random initialization and using the top $r$ singular vectors of $W_0$ as matrix A.
>
> We provide some initial results here (without DP), it can be seen from the plot that matrix $A$ with orthogonal initialization seems to perform slightly better than the existing approach.
> However, the performance gap is not significant enough for a definitive answer.
>
> | Method   | QNLI mean | QNLI variance |
> |---|---|----|
> | Kaiming Init.    | 91.84\%   | 0.38\% |
> | Orthogonal Init. | 92.16\%   | 0.83\%  |
> | SVD Init.    | 91.50\%   | 0.59\%   |
>
> However, although this is a great question with many interesting potential directions, we mainly consider the result of freezing A after initialization, and the choices of initialization is beyond our initial scope of this work. Although all these methods use SGD as the back-bone optimization algorithm, the key ideas and procedures for training are very different from DP. Thus, each of these challenges may deserve an independent project to be investigated. We will mentions these problems as future directions in our revision.

---

> ### Author Response · Authors · 2023-11-19
> **Response to Reviewer E2bX (Part 2)**
>
> > What's the variance in the experiment?
>
> In many of the experiments (especially for RoBERTa), we find that the variance is typically low, and the performance is consistent, therefore we mainly consider the performance. We conducted additional experiments (20 reruns with different seeds) the QNLI experiment, under the same setting as Table 1, and provide the mean and variance of the results here. We note that some rare cases where the algorithm failed to converge has been removed from the statistics.
>
> | Privacy Budget + Method   | QNLI mean | QNLI variance |
> |----|---|---|
> | Non Private + LoRA   | 91.56\%   | 0.43\%  |
> | Non Private + FFA-LoRA    | 91.84\%   | 0.38\%  |
> | $\epsilon = 6$ + LoRA   | 86.55\%   | 1.02\% |
> | $\epsilon = 6$ + FFA-LoRA | 87.42\%   | 0.80\%  |
> | $\epsilon = 3$ + LoRA     | 85.66\%   | 1.20\%  |
> | $\epsilon = 3$+ FFA-LoRA  | 86.18\%   | 1.01\%  |
> | $\epsilon = 1$ + LoRA   | 80.86\%   | 0.28\%     |
> | $\epsilon = 1$+ FFA-LoRA  | 83.63\%   | 1.32\%      |
>
>
> [1] *Yu, Da, Huishuai Zhang, Wei Chen, Jian Yin, and Tie-Yan Liu.* Large scale private learning via low-rank reparametrization. *In International Conference on Machine Learning,PMLR, 2021.*

---

> ### Comment · Reviewer_E2bX · 2023-11-20
> **Reply to Authors**
>
> Thanks for replying.
>
> 1. There still lacks evidence to support the claim "FFA is more suitable for higher noise". I expected the performance gap between FFA-LoRA and vanilla LoRA to increase as the noise magnitude grows, but I did not see such behavior in Tables 1 and 4. Could the authors provide additional numbers or explanations?
>
> 2. Could the authors summarize the goal of the theoretical analysis before diving into the detailed technical claims on $\alpha$ and the full rank property? Also, is there any definition of the equivalent relationship between LoRA and its variance?

---

> ### Author Response · Authors · 2023-11-21
> **Reply to Comment**
>
> We thank the reviewer for their additional comments.
>
> > There still lacks evidence to support the claim "FFA is more suitable for higher noise". I expected the performance gap between FFA-LoRA and vanilla LoRA to increase as the noise magnitude grows, but I did not see such behavior in Tables 1 and 4. Could the authors provide additional numbers or explanations?
>
> As we have stated in the paper, the performance gap between FFA-LoRA and LoRA widens when we are faced with a smaller privacy budget, which was verified in Table 1 and 4. To better illustrate this observation, we provide the following figure on the relationship between injected noise and averaged algorithm accuracy across the 4 tasks. As shown in the figure (<https://imgur.com/a/7FmlPdQ>), as more noise is added, the performance gap widens.
>
> > Could the authors summarize the goal of the theoretical analysis before diving into the detailed technical claims on
>  and the full rank property?
>
> We plan to update the draft such that the theoretical analysis is coherent with our existing statements in the paper. We provide a framework here on how the introduced theorems benefit our arguments made in the paper.
>
> - Our goal is to show that
>     - $\alpha$ does not matter for FFA-LoRA, which is one less hyper parameter to worry about. This claim is backed by the theorem in our paper.
>     - For LoRA, generally big $\alpha$ is good for performance but bad for stability. FFA-LoRA is equivalent to $\alpha = \infty$ for LoRA without being unstable. The equivalence is backed by *theorem 1 in our response* we provide in our response.
> - Many reviewers have mentioned the need for theoretical analysis. We state that the existing literature on the convergence of federated learning can apply to FFA-LoRA, but not LoRA. This is backed by our Theorem on smoothness.
> - The full rank property of $A$ was mentioned in our response to comment on how to construct different methods of initialization. By definition of linear algebra, a full rank $A$ is more expressive that a rank-deficient one, which is why we consider full rank initialization. It is by no means the only approach.
>
> > Also, is there any definition of the equivalent relationship between LoRA and its variance?
>
> The equivalence between LoRA and FFA-LoRA happens when $\alpha \rightarrow \infty$ for LoRA, we provide the formal definition in our theorem on equivalence. Or, informally speaking, we have
>
> $$\lim_{\alpha_{LoRA} \rightarrow \infty} W_{\alpha_{LoRA}}^k = W_{\alpha_{FFA}}^k,$$
>
> where the matrices $W_{\alpha_{LoRA}}^k, W_{\alpha_{FFA}}^k$ are produced by $k$ local iterations of LoRA and FFA-LoRA.
> We want to show that as $\alpha$ increases, LoRA's update on A and B gets closer to the updates made by FFA-LoRA. Which is good since in many cases we want LoRA to have big $\alpha$.

---

> ### Comment · Reviewer_E2bX · 2023-11-22
> **Comments to Authors**
>
> Thanks for replying. Two major issues remain, so I have decided to keep my rating.
>
> 1. There are several noticeable counter-examples in Tables 1&4, and the new figure does not address them. For example, we reach the largest gap between LoRA and FFA-LoRA with $\epsilon=1$ on the QQP and QNLI datasets instead of $\epsilon \in \\{3, 6\\}$. LoRA also outperforms FFA-LoRA by a significant margin on the SST-2 datasets with $\epsilon=1$. These results might be outliers, but we need more investigation before concluding that FFA-LoRA is more robust to DP noise.
>
> 2. The equivalence relationship between FFA-LoRA and LoRA never holds in practice and the benefit of large $\alpha$ is questionable. Theorem 1, in the revision, suggests that a necessary condition for such an equivalence relationship is $\alpha_{LoRA} = \infty$. In practice, $\alpha_{LoRA} \neq \infty$. Indeed, in the provided reference (Kuang et al., 2023), the maximum $\alpha=50.0$ or $128$ if I read the tables in the appendix correctly. Also, tuning $\alpha$ does not seem to impact the accuracy much, as Figure 5(a) in Kuang et al., 2023 shows. Experiments with $\alpha = 0.5$ and $\alpha=50.0$ have almost the same average accuracy.

---

> > ### Author Response · Authors · 2023-11-23
> > **Reply to Comment**
> >
> > We thank the reviewer for their additional comment.
> >
> > 1. As mentioned earlier, the performance is very task-dependent, for a simpler task such as SST-2, it is evident that DP budget does not have a significant affect over accuracy, and this smaller performance gap is more susceptible to outliers, as observed in previous literature (Yu et al., 2021).
> >
> >     After averaging across multiple runs, we can eliminate the outliers and provide a more consistent result. We will add the additional repeated experiments for all performance evaluations shown in Table 1 and 4 in the future.
> >
> > 2. Referencing back to the reference (Kuang et al., 2023) and Table 13 therein, the scaling coefficients used in the experiment ranges from 16 to 128. As the $\alpha$ changes, there are two trends we can see:
> >
> >     - $\alpha = 128$ results in the best possible performance when compared to $\alpha = 16$.
> >     - The optimal learning rate varies for different $\alpha$.
> >     Therefore a grid-search over $\alpha$ and learning rate is needed to get the optimal performance for LoRA. Whereas FFA-LoRA relaxes one of the hyper-parameters.
> >
> >     The reviewer mentioned that "the benefit of large $\alpha$ is questionable". This is indeed true. The best possible $\alpha$ are task-dependent. However, what can be observed is that, in practice a large $\alpha$ always makes the LoRA less stable. When it is needed, FFA-LoRA can serve as a valid alternative when a larger $\alpha$ is preferred.
> >
> >     In addition, for algorithms under iid setting and without consideration of DP, the reviewer claimed that "different choices of $\alpha$ have almost the same average accuracy". We would like to point out that even when this is the case, although the disparity of LoRA is less severe, FFA-LoRA still has on par convergence while only training on half as many parameters.

---

### Official Review · Reviewer_WAsF · 2023-11-01

**Soundness:** 2 fair
**Presentation:** 2 fair
**Contribution:** 2 fair
**Rating:** 5
**Confidence:** 3

**Summary:**

The author proposes FFA-LoRA, a LoRA variant in FL by freezing one of the LoRA weight and training only the other LoRA weight so that it's easy to do model averaging in FL. Empirical results show that FFA-LoRA achieves comparable performance compared with LoRA under different differential privacy guarantees.

**Strengths:**

+ The motivation is sound and the paper writing is easy to follow.
+ Empirical results show competitive performance under different differential privacy and parameter budget.
+ Empirical results are comprehensive, considering multiple tasks and ablation study.

**Weaknesses:**

+ The motivation is straightforward and intuitive, without theoretical insights.

**Questions:**

+ Why rank 16 for MNLI is worse than rank 8 in Table 2?
+ Another intuitive variant is to alternative optimize the two LoRA weights. How would this perform compare with the proposed method?
+ Why non-iid performance is similar to iid performance in Table 3?

---

> ### Author Response · Authors · 2023-11-19
> **Response to Reviewer WAsF**
>
> We are grateful for the reviewer's careful review and constructive comments. Based on your comments, we would like to make the following clarification.
>
> > The motivation is straightforward and intuitive, without theoretical insights.
>
> We appreciate the reviewer's comment on our motivation being straightforward and intuitive. We provide some additional theoretical results regarding the relationship between FFA-LoRA and LoRA, a theorem regarding the convergence properties of FFA-LoRA, and lastly a guarantee on differential privacy. All of which will be added to our revised draft.
> We refer to our "**Response to All**" for a more detailed discussion.
>
> > Why rank 16 for MNLI is worse than rank 8 in Table 2?
>
> We report all our experiments truthfully. It has been shown that even for vanilla LoRA, there is an optimal rank for a specific fine-tuning task, and the performance of LoRA could decrease if rank is too large, the exact optimal rank is task-dependent [1]. This observation can be potentially explained by a similar reason.
>
> We thank the reviewer for the careful observation, and will add statements to address this in the revised version of the paper.
>
> > Another intuitive variant is to alternative optimize the two LoRA weights. How would this perform compare with the proposed method?
>
> This is a great question. We have conducted experiments regarding alternate updates on the weights. The results show slower convergence and no significant performance gains compared to vanilla LoRA.
> Empirically, this means that alternating update is hard to tune: with a smaller learning rate similar to LoRA, it converges slower; with a larger learning rate similar to FFA-LoRA, it is not robust and often unable to converge. For the case of non-iid agents, this algorithm is unable to converge under 5 different random seeds and multiple learning rates $\eta \in \\{0.05, 0.1, 0.2\\}$. (For this experiment, the best LR for LoRA and FFA-LoRA is $\eta=0.05$ and $\eta = 0.5$, respectively.)
> This can be confirmed by both accuracy scores as well as observations of the learning curve during training.
> Additionally, with the consideration of DP, the number of gradient calculations is doubled for alternative update methods, which could result in a higher noise added to ensure privacy. In this case, the drawbacks overshadow the benefits. We will provide a brief discussion on this potential solution in our revision.
>
> We also include one result from the alternate update case here demonstrating its performance in the Table below.
>
> > Why non-iid performance is similar to iid performance in Table 3?
>
> As we stated in Section 5.1, for the heterogeneous setting, we split data based on their labels. Due to the heterogeneity presented to these methods is not strong enough, there is not a strong separation between iid and non-iid experiments for some tasks. Therefore, a performance gap exists between FFA-LoRA and vanilla LoRA, but it is not very significant. To demonstrate the impact of heterogeneity, we included an additional experiment with even stronger heterogeneity.
>
> In the previous MNLI results we ran in the original paper, the non-iid case was formulated using a data label split of $[0.6, 0.2, 0.2], [0.2, 0.6, 0.2], [0.2, 0.2, 0.6]$, which we denote as mild non-iid here. We consider a stronger hetergeneity here, with data label split ratio of $[0.9, 0.05, 0.05], [0.05, 0.9, 0.05], [0.05, 0.05, 0.9]$. The performance degradation is clearly shown, with FFA-LoRA significantly better than LoRA.
>
>
> | Privacy Budget + Method  | MNLI matched | MNLI mismatched |
> |----------------------------|--------------|-----------------|
> | I.I.D. + LoRA              | 86.90\%      | 87.15\%         |
> | I.I.D. + FFA-LoRA          | 87.13\%      | 87.21\%         |
> | I.I.D. + Alt. update       | 86.15\%      | 86.98\%         |
> | Mild Het. + LoRA           | 87.01\%      | 87.33\%         |
> | Mild Het. + FFA-LoRA       | 87.04\%      | 87.36\%         |
> | Severe Het. +  LoRA        | 83.95\%      | 84.51\%         |
> | Severe Het. +  FFA-LoRA    | 86.28\%      | 86.71\%         |
> | Severe Het. +  Alt. update | 35.45\%      | 35.22\%         |
>
> ---
>
> [1] *Zhang, Qingru, Chen, Minshuo, Bukharin, Alexander, He, Pengcheng, Cheng, Yu, Chen, Weizhu, Zhao, Tuo.* Adaptive budget allocation for parameter-efficient fine-tuning. *The Eleventh International Conference on Learning Representation*

---

### Official Review · Reviewer_1Jzn · 2023-11-02

**Soundness:** 3 good
**Presentation:** 4 excellent
**Contribution:** 3 good
**Rating:** 6
**Confidence:** 4

**Summary:**

This paper discuss the potential discordances of applying LoRA in differentially private federated learning: (1) decompose `\DeltaW` to `BA` moves LoRA into a nonlinear regime that potentially cause trouble for aggregation/averaging in model updates (2) the nonlinearity of `BA` cause trouble for DP noise (3) LoRA introduce an extra parameter \alpha. A new algorithm FFA-LoRA is proposed, where instead of updating both `B` and `A` matrices in LoRA, FFA-LoRA only updates the matrix B and keeps A fixed at random initialization.


====== after rebuttal ======

I thank the authors for their response, and would like to maintain the borderline positive score. I really like the idea of fixing one matrix in LoRA, and appreciate the empirical evaluation. I won't provide a stronger support as the experimental setup is a bit unconventional, and it is a bit hard to justify the claims based on the current draft.

**Strengths:**

I like the motivation of the FFA-LoRA algorithm, and appreciate the attempt to provide some analysis on the caveats of LoRA. The experiments on two models (RoBERTa and LLaMA) fine-tuning on a subset of GLUE tasks and a GSM-8K language generation task in both non-DP and DP settings show good empirical performance of FFA-LoRA.

**Weaknesses:**

I thank the authors for providing details of the experimental setup. However, the federated learning setting in experiments seems a bit unconventional with a very small number of clients (only 3 clients). This might be categorized as a cross-silo setting, but it would be good to clearly discuss the targeted application (https://arxiv.org/abs/1912.04977 table 1, https://arxiv.org/abs/2107.06917 section 3.1).

While I appreciate the motivation of analyzing LoRA in section 3, none of the explanations seems to be particularly convincing. The discussion of Discordance (1) and (2) heavily focus on the nonlinear nature of LoRA, but deep neural networks suffer from more severe nonlinearity, it is a bit unclear for me why LoRA `BA` suffers more than multi-layer network `W_1 W_2`. For example, for (1), I believe it not only applies to averaging models from clients, but also to averaging gradients from examples.

I also fail to understand why \alpha becomes an issue for LoRA in  Discordance (3) as it is only a scalar and might potentially be absorbed in learning rate. As shown in table  5, tuning the learning rate helps.

Minor:
The empirical results on local DP seem to be very good with very little accuracy drop compared to non-DP results. It is possible that DP fine-tuning is not a particularly hard task compared to training from scratch, but could the authors share more details about the privacy accounting and important privacy parameters?

FedBert seems to mainly focus on pre-training instead of fine-tuning.

Please cite “Communication-Efficient Learning of Deep Networks from Decentralized Data” for federated learning and the FedAvg algorithm.

**Questions:**

See weakness above.

---

> ### Author Response · Authors · 2023-11-19
> **Response to Reviewer 1Jzn (Part 1)**
>
> We thank the reviewer for their time and effort reviewing our paper. Please see our response below with respect to the specific comments.
>
> > However, the federated learning setting in experiments seems a bit unconventional with a very small number of clients (only 3 clients). This might be categorized as a cross-silo setting, but it would be good to clearly discuss the targeted application
>
> Yes, our focus is indeed the cross-silo setting (similar to existing work in FL LLM, cite some). The major reason is the immense computational burden of LLM training. Also, the difficulties we discussed in this paper are well suited in the context of cross-silo setting, since cross-silo setting is interested in problems such as data-heterogeneity, communication efficiency and privacy, which we were able to address. In our revision, we will clarify with the exact setting.
>
> > While I appreciate the motivation of analyzing LoRA in section 3, none of the explanations seems to be particularly convincing.
>
> We thank the reviewer for acknowledging the motivations of our analysis.
>
> Firstly, to further consolidate our analysis, we have added an additional theorems as well as a corollary. We refer to our "**Response to All**" for these theoretical results. We hope the reviewer find our statements more convincing with the addition of the theoretical results.
>
> Secondly, the disparity we pointed out in the paper is more significant when the weight difference is not-negligible, i.e. $A_1\not\approx A_2$. In terms of averaging gradients from examples, this problem is less of a concern since the learning rate $\eta$ is generally small, and the disparity is not severe. However, for federated learning with averaging models after $K>1$ local update steps, the disparity becomes more problematic.
>
> > fail to understand why $\alpha$ becomes an issue for LoRA in Discordance (3) as it is only a scalar and might potentially be absorbed in learning rate. As shown in table 5, tuning the learning rate helps.
>
> Yes, as we have stated in Section 5.2 of the paper, by jointly tuning learning rate $\eta$ and scaling factor $\alpha$, the fine-tuning algorithm for LoRA is able to converge. However, in vanilla LoRA, as reported by previous works, *tuning learning rate $\eta$ and scaling factor $\alpha$ have very different effects* on the dynamic of the training process as well as the performance, and the exact relationship between them is unknown. In a way, changing both the learning rate and scaling factor is equivalent to tuning the initialization of the random matrices $A$ in the network.
> There are existing works showing that for many tasks (FS-LLM), a bigger scaling factor (as high as 256) is able to converge to higher accuracy, however the algorithm becomes more unstable as $\alpha$ increases, and a re-exploration on the optimal learning rate is needed for a different $\alpha$.
>
> Additionally, *in the context of Differential Privacy*, a higher scaling factor for vanilla LoRA would also change the clipping and noises added to the trainable parameters, which introduces additional difficulties.
>
> On the other hand, for FFA-LoRA, the effect of $\eta$ and $\alpha$ is the same, which means we can set-and-forget one and focus on the other, meaning that we have one less hyper-parameter to tune in experiments.
>
> > Minor: The empirical results on local DP seem to be very good with very little accuracy drop compared to non-DP results. It is possible that DP fine-tuning is not a particularly hard task compared to training from scratch, but could the authors share more details about the privacy accounting and important privacy parameters?
>
> We would like to clarify that no privacy was considered in pre-training, and we only consider the notion of differential privacy in finetuning.
>
> Additionally, the difficulty in DP fine-tuning is an entirely different question. In general, the difficulty of DP training is associated with the dimension of the problem, with higher dimensional data requiring a lower signal-to-noise ratio. Fine-tuning LLMs with privacy was considered to be very difficult due to the high-dimensional trainable parameters.
> However, as pointed out by some recent works on DP fine-tuning, it has been shown that DP fine-tuning empirically exhibits much better performance [1, 2]. This was explained because of the lower intrinsic dimensions within LLMs [3], but the definitive reason for this behavior is largely an open problem.
>
> We have added more theory as well as calculations behind our approach to ensure differential privacy in our "**Response to All**" (Corollary 3), which we reiterate below based on Theorem 1 of [4] and the parallel composition and resistance to post-processing of DP.

---

> ### Author Response · Authors · 2023-11-19
> **Response to Reviewer 1Jzn (Part 2)**
>
> > FedBert seems to mainly focus on pre-training instead of fine-tuning.
>
> The reviewer is correct. In our paper, citation of FedBERT had been referred to as a method for fine-tuning, it is instead intended for pre-training, we will change our statements accordingly.
>
> > Please cite “Communication-Efficient Learning of Deep Networks from Decentralized Data” for federated learning and the FedAvg algorithm.
>
> We will cite this paper in our revision, and we thank the reviewer for the useful suggestion.
>
>
> [1] *Xuechen Li, Florian Tramer, Percy Liang, and Tatsunori Hashimoto.* Large language models can be
> strong differentially private learners. *arXiv preprint arXiv:2110.05679, 2021*
>
> [2] *Da Yu, Huishuai Zhang, Wei Chen, Jian Yin, and Tie-Yan Liu.* Large scale private learning via low-rank reparametrization. *In International Conference on Machine Learning, 2021*
>
> [3] *Xuechen Li, Daogao Liu, Tatsunori B Hashimoto, Huseyin A Inan, Janardhan Kulkarni, Yin-Tat Lee, and Abhradeep Guha Thakurta.* When does differentially private learning not suffer in high dimensions? *Advances in Neural Information Processing Systems, 2022.*
>
> [4] *Martin Abadi, Andy Chu, Ian Goodfellow, H Brendan McMahan, Ilya Mironov, Kunal Talwar, and
> Li Zhang.* Deep learning with differential privacy. *In Proceedings of the 2016 ACM SIGSAC
> conference on computer and communications security*

---

### Author Response · Authors · 2023-11-19
**Response to All**

We thank all the reviewers for their careful evaluations and thoughtful comments.
All reviewers mentioned that our work is well written, our proposed is well motivated and intuitive, and that our provided empirical results demonstrate good performance, on par or better that the state-of-the-art results.
Multiple reviewers have expressed similar concerns regarding aspects of our paper, which we address below.

Multiple reviewers have mentioned that while our motivations are intuitive, our paper is lacking in terms of **theoretical analysis**.
Here, we provide additional analysis regarding several aspects of our paper, including

- Theorem to establish equivalency between LoRA with $\alpha\rightarrow \infty$ and FFA-LoRA.
- Theorem discussing Lipschitz smoothness of FFA-LoRA, which can be used to derive convergence guarantees.
- Careful analysis on the differential privacy guarantee of our federated fine-tuning approach.

These additions will also be added to our revised draft, to appear in either the main paper or the appendix. We hope the reviewers find these additions welcoming.

Many reviewers have raised questions regarding **experimental results** in the paper. We have addressed these questions in our response to each of the reviewers. We mention here that apart from answering the questions, we have also provided additional experiments regarding

- more severe heterogeneity
- different initialization on matrix $A$
- more repeated experiments to better evaluate variance of the experiments
- evaluations of our method on other tasks such as computer vision

We hope that the reviewers find these additional results insightful, and their concerns fully addressed.

---
**The Theoretical results are as follows:**

**Theorem 1 (Scaling Factor):** For local updates with the same initial condition on $\mathbf{W}$, vanilla LoRA update with scaling factor $\alpha\_{LoRA}$ produces trajectory $ \\{ W\_{\alpha\_{LoRA}}^k \\}\_{k\in [K]}$, and FFA-LoRA with scaling $\alpha\_{FFA}$ produces trajectory $\\{W\_{\alpha\_{FFA}}^k\\}\_{k\in [K]}$. Then we have
    $$\lim_{\alpha_{LoRA} \rightarrow \infty} W_{\alpha_{LoRA}}^k = W_{\alpha_{FFA}}^k, \text{ for all } k, \alpha_{FFA}.$$

**Theorem 2 (Smoothness conditions):** Assume that the loss function given weight and dataset is denoted $F(\mathbf{W}, D)$. For a low-rank decomposition on model parameter $\mathbf{W}$ such that $\mathbf{W}(\mathbf{A}, \mathbf{B}) = \mathbf{W}\_0 + \mathbf{B} \mathbf{A}$ satisfying Equation (1). We have the following properties.
1. If $\mathbf{B} $ is trainable, $\mathbf{A}$ is fixed with $||\mathbf{A}|| \leq C$ and $F(\mathbf{W}, D)$ is Lipschitz smooth with respect to $\mathbf{W}$ with factor $L$. The loss function $F(\mathbf{W}(\mathbf{A}, \mathbf{B}))$ is Lipschitz smooth with respect to $\mathbf{B}$ with factor $LC^2$.
2. If both $\mathbf{A}$ and $\mathbf{B}$ are trainable and $F(\mathbf{W}, D)$ is Lipschitz smooth with respect to $\mathbf{W}$ with factor $L$, the loss function $F(\mathbf{W}(\mathbf{A}, \mathbf{B}))$ has no Lipschitz smoothness guarantees with respect to $\[\mathbf{A}, \mathbf{B}\]$.

All the smoothness notions are defined with respect to the matrix Frobenius norm, which is denoted as $||\cdot||$.

**Corollary 3 (Privacy Guarantee):** Given Theorem 1 with moments accountant in [1],  the parallel composition and resistance to post-processing of DP, the mechanism updating FFS-LoRA with locally ran DP-SGD and FedAvg can satisfy $(\epsilon, \delta)$-DP given $\forall i, q=\frac{|B_i|}{|N_i|}$, the number of total local updates $T$ of each client and  $\sigma = O(\frac{q\sqrt{T\log(1/\delta)}}{\epsilon})$.

(The exact $\sigma$ is computed by the Pytorch's Opacus package [2] numerically given $q, T, \epsilon, \delta$).

---
**We attach the detailed analysis in the comment, due to the space constraint of official comments.**

[1] *Martin Abadi, Andy Chu, Ian Goodfellow, H Brendan McMahan, Ilya Mironov, Kunal Talwar, and
Li Zhang.* Deep learning with differential privacy. *In Proceedings of the 2016 ACM SIGSAC
conference on computer and communications security*


[2] <https://github.com/pytorch/opacus>

---

> ### Author Response · Authors · 2023-11-19
> **Proof for Theorem 1**
>
> **Theorem 1 (Scaling Factor):** For local updates with the same initial condition on $\mathbf{W}$, vanilla LoRA update with scaling factor $\alpha\_{LoRA}$ produces trajectory over $k$ local iterations $ \\{ W\_{\alpha\_{LoRA}}^k \\}\_{k\in [K]}$, and FFA-LoRA with scaling $\alpha\_{FFA}$ produces trajectory $\\{W\_{\alpha\_{FFA}}^k\\}\_{k\in [K]}$. Then we have
>     $$\lim_{\alpha_{LoRA} \rightarrow \infty} W_{\alpha_{LoRA}}^k = W_{\alpha_{FFA}}^k, \text{ for all } k, \alpha_{FFA}.$$
>
> **Proof:** The theorem starts with initial condition on $\mathbf{W}$, since $\mathbf{W} = \mathbf{W}\_0 + \alpha \mathbf{B}\_\alpha \mathbf{A}\_\alpha$ and that the initialization of $\mathbf{A}$ is non-zero, this condition implies that $\mathbf{A}\_\alpha = \mathbf{A}\_1 = \mathbf{A}$, and $\mathbf{B}\_\alpha = \frac{1}{\alpha}\mathbf{B}\_1$. Now we compare the update of the two algorithms given the same initial conditions.
>
> From Theorem 1 we know that for FFA-LoRA, different $\alpha_{FFA}$ does not affect its dynamics, without loss of generality, we consider the case where $\alpha_{FFA} = 1$.
>
>
> The FFA-LoRA update is as follows, the only update is on $B$:
>
> \begin{align*}
>         W^{k+1}\_{FFA} = W\_0 + 1 \times B^{k+1}A^k = W\_0 + (B^k - \eta \nabla B^k)A^k = W^k  - \eta \nabla B^k A^k
> \end{align*}
> The rest of the proof is given by induction, as long as the limit holds for the $k+1$-th local iteration given that the $k$-th iteration holds.
> Without the loss of generality, we first consider when $\alpha_{LoRA} = 1$, then for iteration $k$, we denote the learning rate as $\eta_1$, denote the matrices and their gradient as $\mathbf{A}^k_{1}, \mathbf{B}^k_1$ and $\nabla \mathbf{A}^k_{1}, \nabla \mathbf{B}^k_{1}$, respectively. And by definition, we have the update that
>
> \begin{align*}
>         \mathbf{A}^{k+1}\_1 &\leftarrow \mathbf{A}^k\_{1}-\eta\_1 \nabla \mathbf{A}^k\_{1}\\\\
>         \mathbf{B}^{k+1}\_{1} &\leftarrow \mathbf{B}^k\_{1}-\eta\_1 \nabla \mathbf{B}^k\_{1}
> \end{align*}
> And the update of the original weight matrix $W$ becomes
>     \begin{align*}
>         W^{k+1}\_{1} = W\_0 + \Delta W^{k+1}\_{1} &=  W_0 + (\mathbf{B}^k\_{1}-\eta_1 \nabla \mathbf{B}^k\_{1})(\mathbf{A}^k\_{1}-\eta\_1 \nabla \mathbf{A}^k\_{1})\\\\
>         &= W^k\_1 - \eta\_1 \left(\nabla \mathbf{B}^k\_{1}\mathbf{A}^k\_{1} +  \mathbf{B}^k\_{1}\nabla\mathbf{A}^k\_{1} \right) + \eta^2 \nabla\mathbf{B}^k\_{1}\nabla\mathbf{A}^k\_{1}
>     \end{align*}
> Since LoRA do not satisfy the conditions provided in Theorem 1, changing $\alpha\_{LoRA}$ will affect its updates. When we choose a different $\alpha\_{LoRA} = \alpha$ and corresponding $\eta\_{\alpha} = \frac{\eta\_1}{\alpha^2}$, we can write the update of LoRA as
>
> \begin{align*}
>         \mathbf{A}^{k+1}\_\alpha &\leftarrow \mathbf{A}^k\_{\alpha}-\eta\_\alpha \nabla \mathbf{A}^k\_{\alpha}
>         = \mathbf{A}^k\_\alpha-\frac{\eta\_1}{\alpha}\nabla \mathbf{A}^k\_\alpha
>         =\mathbf{A}^k\_1-\frac{\eta\_1}{\alpha}\nabla \mathbf{A}^k\_1\\\\
>         \mathbf{B}^{k+1}\_\alpha &\leftarrow \mathbf{B}^k\_\alpha-\eta\_\alpha \nabla \mathbf{B}^k\_\alpha
>         = \mathbf{B}^k\_\alpha-\frac{\eta\_1}{\alpha}\nabla \mathbf{B}^k\_\alpha
>         = \frac{1}{\alpha}\mathbf{B}^k\_1-\frac{\eta\_1}{\alpha}\nabla \mathbf{B}^k\_1\\\\
>         W^{k+1}\_\alpha &=  W\_0 + \alpha
>         (\frac{1}{\alpha}\mathbf{B}^k\_1-\frac{\eta\_1}{\alpha}\nabla \mathbf{B}^k\_1)
>         (\mathbf{A}^k\_1-\frac{\eta\_1}{\alpha}\nabla \mathbf{A}^k\_1)\\\\
>         &=  W_0 + \mathbf{B}^k\_1\mathbf{A}^k\_1 - \eta\_1 \nabla \mathbf{B}^k\_1 \mathbf{A}^k\_1 - \frac{\eta\_1}{\alpha} \mathbf{B}^k\_1 \nabla \mathbf{A}^k\_1 - \frac{\eta\_1^2}{\alpha} \nabla \mathbf{B}^k\_1 \nabla \mathbf{A}^k\_1
> \end{align*}
> Therefore we have
> \begin{align*}
>         \lim\_{\alpha\_{LoRA}\rightarrow \infty} W\_{\alpha\_{LoRA}}^{k+1} &=
>         \lim\_{\alpha \rightarrow \infty} W\_0 + \mathbf{B}^k\_1\mathbf{A}^k\_1 - \eta_1 \nabla \mathbf{B}^k\_1 \mathbf{A}^k\_1 - \frac{\eta\_1}{\alpha} \mathbf{B}^k\_1 \nabla \mathbf{A}^k\_1 - \frac{\eta\_1^2}{\alpha} \nabla \mathbf{B}^k\_1 \nabla \mathbf{A}^k\_1\\\\
>         &= W^k\_1 - \eta\_1 \nabla \mathbf{B}^k\_1 \mathbf{A}^k\_1\\\\
>         &= W^{k+1}\_{FFA}
> \end{align*}
> Which completes our proof.

---

> ### Author Response · Authors · 2023-11-19
> **Proof for Theorem 2**
>
> **Theorem 2 (Smoothness conditions):** Assume that the loss function given weight and dataset is denoted $F(\mathbf{W}, D)$. For a low-rank decomposition on model parameter $\mathbf{W}$ such that $\mathbf{W}(\mathbf{A}, \mathbf{B}) = \mathbf{W}\_0 + \mathbf{B} \mathbf{A}$ satisfying Equation (1). We have the following properties.
> 1. If $\mathbf{B} $ is trainable, $\mathbf{A}$ is fixed with $||\mathbf{A}|| \leq C$ and $F(\mathbf{W}, D)$ is Lipschitz smooth with respect to $\mathbf{W}$ with factor $L$. The loss function $F(\mathbf{W}(\mathbf{A}, \mathbf{B}))$ is Lipschitz smooth with respect to $\mathbf{B}$ with factor $LC^2$.
> 2. If both $\mathbf{A}$ and $\mathbf{B}$ are trainable and $F(\mathbf{W}, D)$ is Lipschitz smooth with respect to $\mathbf{W}$ with factor $L$, the loss function $F(\mathbf{W}(\mathbf{A}, \mathbf{B}))$ has no Lipschitz smoothness guarantees with respect to $\[\mathbf{A}, \mathbf{B}\]$.
>
> All the smoothness notions are defined with respect to the matrix Frobenius norm, which is denoted as $||\cdot||$.
>
> **Proof:**
> First we show that, given $\mathbf{W}(\mathbf{A}, \mathbf{B}) = \mathbf{W}_0 + \mathbf{B} \mathbf{A}$,  and the gradient on $\mathbf{W}$ is denoted as $\nabla_W F$, then we can write the gradients on matrix $\mathbf{B}$ as $\nabla_B F = \nabla_W F \mathbf{A}^T$, since
> \begin{align*}
> \langle \mathbf{B}_1 - \mathbf{B}_2, \nabla_B F\rangle & = \langle \mathbf{W}(\mathbf{A}, \mathbf{B}_1) - \mathbf{W}(\mathbf{A}, \mathbf{B}_2), \nabla_W F\rangle\\\\
> &= \langle  \mathbf{B}_1 \mathbf{A} - \mathbf{B}_2 \mathbf{A}, \nabla_W F\rangle\\\\
> &= \langle  \mathbf{B}_1 - \mathbf{B}_2 , \nabla_W F \mathbf{A}^T\rangle
> \end{align*}
> Similarly, we have $\nabla_A F = \mathbf{B}^T \nabla_W F$. Using the gradients on $\mathbf{A}$ and $\mathbf{B}$, we provide the proof for all the properties.
>
> 1. For property 1, we know that for any given $\mathbf{B}_1, \mathbf{B}_2$,
> \begin{align*}
>  & ||\nabla\_{B} F(\mathbf{W}(\mathbf{A}, \mathbf{B}\_1)) - \nabla\_{B} F(\mathbf{W}(\mathbf{A}, \mathbf{B}\_2))|| \\\\
>  =& ||\nabla\_{W} F(\mathbf{W}(\mathbf{A}, \mathbf{B}\_1)) \mathbf{A}^T - \nabla\_{W} F(\mathbf{W}(\mathbf{A}, \mathbf{B}\_2)) \mathbf{A}^T|| \\\\
>  \leq& L ||\mathbf{W}( \mathbf{A}, \mathbf{B}\_1)  - \mathbf{W}(\mathbf{A}, \mathbf{B}\_2) || \times ||\mathbf{A}||\\\\
>  \leq& L || \mathbf{B}\_1  -  \mathbf{B}\_2 ||  ||\mathbf{A}||^2 \\\\
>  \leq& L C^2||\mathbf{B}\_1  -  \mathbf{B}\_2 ||
> \end{align*}
>
> 2. For the second property, for the ease of notation, we introduce the stacked variable $\mathbf{x} := [\mathbf{A}, \mathbf{B}]$. We construct a counter-example such that the function is not Lipschitz smooth with respect to $\mathbf{x}$.
>
> We consider $\mathbf{W}, \mathbf{A}, \mathbf{B} \in \mathbb{R}^{d\times d}, F(\mathbf{W}) = \frac{1}{2}||\mathbf{W}||^2$ with $\mathbf{W}\_0 = 0$. Then we consider a sequence $\\{\mathbf{x}\_k\\}\_{k \in \mathbb{N}}$ such that $\mathbf{x}\_k = [\mathbf{A}\_k, \mathbf{B}\_k] = [k \mathbf{I}\_d, k \mathbf{I}\_d]$, then
>
> \begin{align*}
> & \lim\_{k \rightarrow \infty} \frac{||\nabla\_x \mathbf{W}(\mathbf{A}\_k, \mathbf{B}\_k) - \nabla\_x \mathbf{W}(\mathbf{A}\_0, \mathbf{B}\_0)||}{||\mathbf{x\_k} - \mathbf{x\_0}||} \\\\
> =& \lim\_{k\rightarrow \infty} \frac{||\nabla\_A \mathbf{W}(\mathbf{A}\_k, \mathbf{B}\_k) - \nabla\_A \mathbf{W}(\mathbf{A}\_0, \mathbf{B}\_0)|| + ||\nabla\_B \mathbf{W}(\mathbf{A}\_k, \mathbf{B}\_k) - \nabla\_B \mathbf{W}(\mathbf{A}\_0, \mathbf{B}\_0)||}{||\mathbf{A}\_k - \mathbf{A}\_0|| + ||\mathbf{B}\_k - \mathbf{B}\_0||}\\\\
> =& \lim_{k\rightarrow \infty} \frac{||k^3 \mathbf{I}_d|| + |k^3 \mathbf{I}_d||}{||k \mathbf{I}_d|| + ||k \mathbf{I}_d||}\\\\
> =& \infty
> \end{align*}
>
> For this example, we can see that although $F(\mathbf{W})$ is 1-Lipschitz smooth, the function is not smooth with respect to $\mathbf{x}$.

---

> ### Author Response · Authors · 2023-11-19
> **Proof for Corollary 3**
>
> **Corollary 3 (Privacy Guarantee):** Given Theorem 1 with moments accountant in [1],  the parallel composition and resistance to post-processing of DP, the mechanism updating FFS-LoRA with locally ran DP-SGD and FedAvg can satisfy $(\epsilon, \delta)$-DP given $\forall i, q=\frac{|B_i|}{|N_i|}$, the number of total local updates $T$ of each client and  $\sigma = O(\frac{q\sqrt{T\log(1/\delta)}}{\epsilon})$.
>
> (The exact $\sigma$ is computed by the Pytorch's Opacus package [2] numerically given $q, T, \epsilon, \delta$).
>
> **Proof:**
> Firstly, we consider the local datasets $\{D_i\}_{i\in [n]}$for the FL network to be disjoint chunks of the global dataset.
> The DP-SGD with FedAvg used in our paper to train LoRA or FFA-LoRA can be considered as
> - (A) locally updating trainable parameters with DP-SGD,
> - (B) averaging the trainable parameters from clients on the server, and
> - (C) repeating the above two steps for some iterations.
>
> The privacy loss of (A) can be composed by moment accountants used in [1].
> The privacy loss of all clients performing local updates can be composed by the parallel composition property of DP.
> The averaging on the server in (B) is a post-processing operation that does not introduce privacy loss.
> Privacy loss of multiple FL rounds of (C) can again be composed with moment accountants used in [1].
> Eventually, we can convert the moment accountants to $(\epsilon, \delta)$-DP as Theorem 1 in [1].
>
>
> [1] *Martin Abadi, Andy Chu, Ian Goodfellow, H Brendan McMahan, Ilya Mironov, Kunal Talwar, and
> Li Zhang.* Deep learning with differential privacy. *In Proceedings of the 2016 ACM SIGSAC
> conference on computer and communications security*
>
>
> [2] <https://github.com/pytorch/opacus>

---

### Meta-Review · Area_Chair_9Rp4 · 2023-12-12

**Metareview:**

(a) Summarize the scientific claims and findings of the paper based on your own reading and characterizations from the reviewers.

The author proposes FFA-LoRA, a LoRA variant in FL by freezing one of the LoRA weight and training only the other LoRA weight so that it's easy to do model averaging in FL. Empirical results show that FFA-LoRA achieves comparable performance compared with LoRA under different differential privacy guarantees.

(b) What are the strengths of the paper?

 FFA-LoRA algorithm is well motivated. The empirical gains are clear in the main experiments. Empirical results are comprehensive, considering multiple tasks and ablation study.

(c) What are the weaknesses of the paper? What might be missing in the submission?

The federated learning setting in experiments seems a bit unconventional with a very small number of clients (only 3 clients). The explanations and the analysis of LoRA are not convincing. The discussion of Discordance (1) and (2) heavily focus on the nonlinear nature of LoRA, but deep neural networks suffer from more severe nonlinearity.

 The performance gap between the vanilla LoRA and the proposed FFA-LoRA remains the same across various privacy budgets, including \epsilon=0. Such an empirical result suggests that the impact of DP noise is the same on both the vanilla LoRA and the proposed FFA-LoRA.

**Justification For Why Not Higher Score:**

The experimental setup is a bit unconventional, and it is a bit hard to justify the claims based on the current draft.

**Justification For Why Not Lower Score:**

The simple but interesting new idea of FFA-LoRA is tested on various FL tasks and shown to give improved performance. This empirical investigation is appreciated by the reviewers.

---

### Decision · Program_Chairs · 2024-01-16

Accept (poster)